# Application of Reinforcement Learning and Deep Learning in Multiple-Input and Multiple-Output (MIMO) Systems

**DOI:** 10.3390/s22010309

**Published:** 2021-12-31

**Authors:** Muddasar Naeem, Giuseppe De Pietro, Antonio Coronato

**Affiliations:** Institute of High Performance Computing and Networking, National Research Council of Italy, 80131 Napoli, Italy; giuseppe.depietro@icar.cnr.it (G.D.P.); antonio.coronato@icar.cnr.it (A.C.)

**Keywords:** reinforcement learning, deep learning, MIMO systems, signal, channel estimation, detection communication, BS, positioning, localization, CSI, resource allocation, mmWave communication

## Abstract

The current wireless communication infrastructure has to face exponential development in mobile traffic size, which demands high data rate, reliability, and low latency. MIMO systems and their variants (i.e., Multi-User MIMO and Massive MIMO) are the most promising 5G wireless communication systems technology due to their high system throughput and data rate. However, the most significant challenges in MIMO communication are substantial problems in exploiting the multiple-antenna and computational complexity. The recent success of RL and DL introduces novel and powerful tools that mitigate issues in MIMO communication systems. This article focuses on RL and DL techniques for MIMO systems by presenting a comprehensive review on the integration between the two areas. We first briefly provide the necessary background to RL, DL, and MIMO. Second, potential RL and DL applications for different MIMO issues, such as detection, classification, and compression; channel estimation; positioning, sensing, and localization; CSI acquisition and feedback, security, and robustness; mmWave communication and resource allocation, are presented.

## 1. Introduction

The main problem with the current wireless communication infrastructure is its dependence on either increasing the spectrum or densifying the cells to obtain the targeted area throughput. Unfortunately, such resources are scarce and are approaching their saturation level in near future. Moreover, increasing the spectrum or densifying the cells increases the hardware price and latency. The spectral efficiency, which can enhance the area throughput, has remained essentially unchanged during the fast development in the wireless systems. Therefore, a wireless access technology must improve the wireless area throughput without increasing the spectrum or densifying the cell to fulfill the essentials requirements of the wireless carriers.

MIMO is the most promising and fascinating wireless access technology that is able to deliver the requirements of 5G and beyond networks. The performance of communication systems has been enhanced thanks to the use of MIMO schemes. MIMO utilizes many dimensions that account for multiple antennas, multiple users, and time and frequency resources. Multi-User MIMO (MU-MIMO) is a MIMO system with one BS equipped with many antennas and provides service to more than one downlink user in a one-time slot. Massive MIMO (Ma-MIMO) further extends MIMO systems by using hundreds and thousands of antennas at BS to enhance throughput and spectral efficiency. MIMO technology is integrating bandwidth, radios, and antennas to achieve higher speed as well as capacity for the incoming 5G [1]. The ability of Ma-MIMO, in particular, to improve spectral efficiency and throughput has turned it a powerful technology for emerging wireless standards [2,3].

Recent advances in AI and ML, specifically the success of RL [4] and DL, have brought many significant applications and advancement in different research areas including robotics and autonomous control [5,6,7,8], communication and networking [9,10,11,12], natural language processing [13,14,15,16], games and self-organized systems [17,18,19,20], autonomous IoT [21,22,23,24,25,26], scheduling management and configuration of resources [27,28,29,30], computer vision [31,32,33,34], and healthcare [35,36,37]. Embedding AI technologies like DRL into the 5G mobile, MIMO communication, and wireless communication systems is well justified [38]. More precisely, data produced by mobile environments are increasingly heterogeneous due to their collection from many sources with different formats, and they have complex correlations.

In the areas of MIMO systems, recently RL and DL have been applied as an emerging solution to address many challenges efficiently. These technologies have introduced many solutions to different aspects of MIMO communication such as signal detection, classification, and compression; positioning, sensing, and localization; security and robustness; mmWave communications; and resource allocation. AI-enabled MIMO communication is an optimal tool that can give wireless systems the flexibility, intelligence, and efficiency needed to handle the scarce radio resource efficiently and enable top quality of service to the customers. In our work, we are making a research contribution by providing a comprehensive overview of the potential applications of RL, DL, and the mixture of both in different areas of MIMO communication.

The remaining of the paper is organized as follows. Section 2 presents a brief introduction to RL, DL, and MIMO technology. Section 4 presents a comprehensive application of RL and DL in different areas of MIMO communication. Next, we present statistics and impacts in Section 5, followed by a discussion in Section 6. Section 3 overviews some recent survey papers. This survey is concluded in Section 7.

## 2. Technical Background

This section provides a brief introduction to RL, DL, and MIMO systems.

### 2.1. Reinforcement Learning

RL is a sub-field of Machine Learning where an agent interacts with an environment to achieve a goal, and learning takes place interaction-after-interaction. This section introduces some basic mechanisms and terminology. A detailed presentation of RL can be found in [39].

In RL, an **Agent** is an entity (algorithm/robot/player, etc.) that interacts with a given environment (problem/smart space/game, etc.) by performing actions, and receives a feedback (penalty/reward) from the environment after any action selected as described in Figure 1. The reward is the mechanism that enables the agent to understand whether the action selected has produced a positive or a negative effect concerning the final goal.

A **Policy** is a strategy that indicates to the agent which action to select in every state of the environment. The agent has to learn the optimal policy, that is, the one that maximizes the cumulative reward over the long run.

A RL problem is defined as a **MDP**. A MDP is a tuple $\left(S,A,P_{a},R_{a},\gamma \right)$, where *S* is a set of states; *A* is a set of actions; $P_{a}=P_{r}\left(s_{t+1}=s^{\prime }|s_{t}=s,a_{t}=a\right)$ is the transition probability (i.e., the probability of achieving $s^{\prime }$ at time t+1, after having selected *a* in *s* at time *t*), $R_{a}\left(s,s^{\prime }\right)$ is the expected reward or immediate reward obtained when transitioning from state *s* to state $s^{\prime }$ when action *a* was taken, respectively; and γ is a discount factor.

RL schemes are normally categorized into two major types: model-free and model-based algorithms. Model-based RL algorithms need a precise description of the dynamics of the environment in terms of the state-transition probability distribution. These methods (e.g., DP) compute the optimal policy by solving systems of equations. Whereas, model-free RL techniques are adopted when there is not a precise description of the model or its solution would be too complicated. This class of algorithms interacts directly with the environment (or with an emulator of it) using Trial&Error schemes to learn the optimal policy. In inverse RL [40,41,42], we study an agent’s objectives, values, or rewards with the help of employing insights of its behavior. Several methods are available (e.g., MMC, TD, etc.). An overview of such methods is reported in [43], whereas a guideline useful to help to choose the algorithm depending on the kind of problem is defined in [35].

### 2.2. Deep Learning

DL has revolutionized many research areas with its ability to learn better models from huge volumes of data [44]. Such technology relies on a new generation of Artificial Neural Networks (ANNs) called Deep Neural Networks (DNNs). This subsection presents a brief overview of DL techniques before approaching the next section.

The premises is that the performance of a DNN is generally superior to the one of a classic ANN at the cost of greater training time that, however, can be reduced by using advanced hardware (e.g., GPU) and/or special techniques (e.g., Transfer Learning) [45]. The design of a DNN is crucial for success. We start this subsection by discussing some of the most used DL architectures.


**Convolutional Neural Networks**
Convolutional Neural Networks (CNNs) are ANNs with a much higher number of layers and nodes. They are typically adopted for images classification. An CNN needs less preprocessing as compared to other classification schemes. Relevant filters are used in CNNs to capture the temporal and spatial dependencies in the image [46,47]. The most common CNN architectures are ZFNet, ResNet, GoogleNet, VG-GNet, AlexNet, and LeNet [48].
**Recurrent Neural Networks**
RNN is another important architecture for DL. Differently from CNN, where the layers are sequentially connected, in a RNN there are some nodes whose output is reported back in the input of a previous node. In this way, the network is capable of remembering some information time-related. Recurrent Neural Networks (RNNs), indeed, are massively applied for time-series analysis and prediction in a configuration called LSTM [49].
**Generative Adversarial Networks**
A Generative Adversarial Network (GAN) consists of two sub-networks—the discriminator and the generator, where the later produces the content and the former validates it. GAN adopts feed-forward and relies typically on CNNs [50].
**Deep Belief Networks**
Deep Belief Networks (DBNs) are generative neural networks with undirected connections between some layers called Restricted Boltzmann Machines. These layers can be trained using a very fast unsupervised learning algorithm called Contrastive Divergence. In DBNs, hidden patterns are learned globally, while in every layer of other deep nets complex patterns are learned progressively [51].
**Autoencoders**
Autoencoders are applied to reduce the dimension of data and to detect problems. The first layer in an autoencoder is an encoding layer, whereas the transpose of it is used as a decoder. Training is unsupervised and the Regression/Classification problems may be addressed and optimized using Stochastic gradient descent. Input data are translated to a latent space denoted by the encoder, as given below:
(1)h=f(x)Input data are reconstructed from the latent space denoted by the decoder as described below.
(2)r=g(h)Autoencoders can be represented essentially by the below equation. *r* is the decoded output and it will be identical to input *x*
(3)g(f(x))=r
**Radial Basis Function Neural Networks**
RBFNN is a type of ANN that utilizes radial basis functions (RBF) as activation functions. The output of the RBFNN is a linear combination of RBF of the neuron parameters and inputs. RBNN has only one hidden layer which is known as a feature vector. Training in RBNN is faster than in MLP but classification in RBNN takes more time than MLP.

### 2.3. MIMO Communication

MIMO is a wireless technology that employs multiple antennas at transmitters and receivers sides to communicate more data simultaneously as can be seen in Figure 2. MIMO are supported by all wireless devices compliant to 802.11n standard. This subsection briefly reviews MIMO technology and its different schemes.

MIMO communication uses a multi-path, which is a natural radio-wave phenomenon. Multi-path-transmitted signals may bounce off objects, like ceilings, walls, etc., and arrive at the receiver multiple times at different times and angles. Before the inception of MIMO, interference occur due to multi-path, which can slow down the communication. MIMO technology with multi-path, however, uses smart transmitters and receivers with the addition of spatial dimension that grants enhanced performance and range.

MIMO improves signal-capturing of the receiver by empowering antennas to combine signals coming from multiple paths at different times. Smart antennas get the benefit of spatial diversity technique that makes surplus antennas useful. The antennas can increase the range by adding receiver diversity when outnumbered spatial streams.


**Single User MIMO**
Single user MIMO, or multi-antenna MIMO, has more than one antenna both at the transmitter side and receiver side. There are some special variants of MIMO such as “Multiple-input single-output” (MISO) (only one antenna at the receiver side), “Single-input multiple-output” (SIMO) (single antenna at the transmitter side), and a special scenario when transmitter as well as receiver have one antenna is called SISO [52].
**Multi-user MIMO**
Multi-User MIMO MU-MIMO has been considered in recent WiMAX and 3GPP standards as a candidate technology by various companies such as Freescale, Nokia, Philips, Huawei, TI, Ericsson, Qualcomm, Intel, and Samsung. MU-MIMO systems are more suitable for low-complexity mobile phones with a few receiving antennas, while single-user MIMO’s are more suitable for complex devices having many antennas due to their higher per-user throughput. Moreover, enhanced MU-MIMO uses advanced precoding and decoding techniques.
**Cooperative MIMO (CO-MIMO)**
Cooperative MIMO (CO-MIMO) employs multiple surrounding BS to jointly transmit and receive signals to and from users. This prevents inter-cell interference in neighboring BS as may be experienced with traditional MIMO systems.
**Macrodiversity MIMO**
Macrodiversity MIMO is a type of space diversity approach that applies many transmit/receive BS for coherent communication with single/multiple users. It is possible that users are distributed in a coverage area that has the same resources of time and frequency [53,54,55]. The transmitters, as well as the users in multi-user microdiversity MIMO, are far apart as compared to that of conventional microdiversity MIMO approaches (e.g., SU-MIMO). As a result, each constituent connection in the virtual MIMO connection has a unique average link SNR. Macrodiversity MIMO techniques face some theoretical and practical challenges. One of the most fundamental issues is to get knowledge about how aggregated system capacity is affected by different average link SNRs and the performance of users individually in fading environments [56].
**Massive MIMO**
Massive MIMO Ma-MIMO is a scheme where the number of terminals is inferior to of BS antennas [57]. The maximum benefits of the massive MIMO in a rich scattering environment can be obtained by applying simple beamforming schemes such as zero-forcing (ZF), maximum ratio combining (MRC) [58], or maximum ratio transmission (MRT) [59]. However, it is difficult to achieve these advantages without the availability of accurate CSI.

## 3. Related Survey Papers

This section will review some recent related surveys, their contribution, and limitations as well as the contribution of our work.

The list of related survey papers is shown in Table 1. The work in [60] considers the area of wireless networks and compares application of three DRL sub-techniques: DDPG, NEC, and VBC for optimization. More precisely, the comparison of three DRL approaches is carried out on experiments being performed on a real-world network operation dataset. Although authors have performed extensive experiments, they limited their analysis to three methods only and very little knowledge can be extracted on the application of DRL in MIMO systems.

Another survey paper [61] focused on the radio-resource allocation in multi-cell networks via DL. Authors have also compared methods qualitatively, in terms of their data training and techniques, inputs/outputs, and objectives. A supervised DL design is presented as a solution to power allocation and sub-band problems in a multi-cell network.

The mmWave communication is discussed in [62,63] using channel estimation and signal processing methods, respectively. The former work comprehensively reviews the state-of-the-art channel estimation methods linked to different mmWave system frameworks. Ma-MIMO mmWave was also considered, but without the use of RL and DL methods. The Ma-MIMO wave aspect is also considered in [63], but it mainly reviews challenges of signal processing in mmWave wireless systems with a particular concern for higher carrier frequencies MIMO communication.

An encyclopedic overview of the application of DL in wireless and mobile networking is presented in [64]. The authors have bridged the research gap between DL and wireless and mobile networking by highlighting the crossovers between these two areas. However, the area of the MIMO system (i.e., the focus of our current work) was not appropriately discussed.

The work in [65] presents a comprehensive state-of-the-art on ML based link quality estimators generated from empirical data. The ML-based link quality estimation architectures are analyzed and existing open-source datasets are also reviewed.

The Ma-MIMO systems are highlighted in [66] while presenting the enabling techniques needed for 5G and 6G architecture. The fundamental challenges associated with signal detection, energy efficiency, user scheduling, precoding, channel estimation, and pilot contamination in a Ma-MIMO communication are discussed. The authors have also outlined visible light communication, ultra Ma-MIMO, terahertz communication, as well as ML and DL for Ma-MIMO systems, but they did not consider other areas of MIMO communication.

A survey paper on DRL techniques that was proposed to solve emerging problems in communications and networking is presented in [67]. The authors have addressed issues such as connectivity preservation, network security, data offloading, data rate control, wireless caching, and dynamic network access. Moreover, DRL applications for data collection, resource sharing, and traffic routing are discussed. However, MIMO communication was partially discussed in few subsections. Similarly, the work in [68] presents an overview of array signal processing techniques for Ma-MIMO communication.

An overview of the DL-based cybersecurity applications for mobile and wireless networks is presented in [69]. The authors have addressed cybersecurity features like privacy preservation, software attacks, attacks, and infrastructure threads.

Different cases of 5G wireless communication including cybersecurity, energy efficiency, caching, Ma-MIMO, channel coding, and modulation classification based on AI techniques are discussed in [70]. The authors were interested in more general AI-based applications for 5G wireless communication.

Although there are few review papers related to applications of ML, RL, and DL reported in Table 1, they do not discuss the applications of RL and DL for MIMO communications. There is a need for a review that particularly outline useful applications of RL and DL for different aspects of MIMO communication.

In this paper, we have presented a comprehensive state-of-the-art on the application of RL and DL in different aspects of MIMO communication such as detection, classification, and compression; channel estimation; positioning, sensing, and localization; CSI acquisition and feedback, security and robustness; mmWave communication; and resource allocation. We have also listed the contribution of some survey papers, their limitations for MIMO communication areas, and our contribution in Table 1.

**Table 1 sensors-22-00309-t001:** List of related surveys.

Paper	Technology	Year	Area	Contribution	Limitation
[60]	DRL	2020	Wireless Network Optimization	Only three DRL techniques: DDPG, NEC, and VBC, are considered for wireless network optimization. Their performances are compared in terms of rate and convergence speed improvement.	Only three DRL methods are taken into account without concerning about MIMO aspects.
[62]	Channel Estimation Techniques	2020	mmWave communication	Review of the channel estimation methods associated with the different mmWave system architectures	Only one area of MIMO communication (i.e., mmWave) is discussed, as well as DL and RL techniques are not considered.
[63]	Signal Processing	2016	mmWave Ma-MIMO communication	Survey of signal processing challenges in mmWave systems, especially focusing on issues due to utilizing MIMO communication at higher carrier frequencies.	Only mmWave communication with signal processing techniques are discussed, as well as DL and RL techniques are not considered.
[61]	DL	2019	Multi-cell networks	Review the application of DL for the radio resource allocation in multi-cell networks.	Focused only on resource allocation.
[64]	DL	2019	Mobile and Wireless Networking	Application of DL in mobile and wireless networking	MIMO systems are not considered.
[65]	ML	2021	Link Quality Estimation	Review ML-based link quality estimation models. It addresses quality requirements and standard design steps perspectives using performance data.	General ML techniques are concerned.
[66]	—	2020	Ma-MIMO	Presents fundamental challenges related to signal detection, energy efficiency, user scheduling, precoding, channel estimation and pilot contamination in a Ma-MIMO system, and solutions to these challenges.	General aspects of Ma-MIMO are considered without application of DL and RL.
[67]	DRL	2019	Communications and Networking	Connectivity preservation, network security, data offloading, data rate control, wireless caching, and dynamic network access issues are addressed.	MIMO systems are not discussed in detail.
[69]	DL	2021	Cybersecurity in Mobile Networks	Cybersecurity aspects: privacy preservation, software attacks, attacks and infrastructure threads are discussed.	No MIMO application.
[70]	AI	2020	5G Wireless Systems	An in-depth review of AI for 5G wireless communication systems including cyber-security, network management, and radio resource allocation	Only Ma-MIMO were discussed in one subsection using general AI approaches.
[68]	Array Signal Processing Techniques	2019	Enhanced Massive MIMO	A review of array signal processing in Ma-MIMO communications.	Only Ma-MIMO systems are considered with array signal processing techniques. No application of DL in MIMO.
Our work	RL and DL	2022	MIMO communication	Comprehensive overview of the application of RL and DL in different aspects of MIMO communication.	The tutorial aspect of our survey only presents a brief introduction to RL and DL.

## 4. RL and DL Application in MIMO

This section presents a comprehensive review of the applications of DL and RL in different areas of MIMO.

### 4.1. Detection, Classification, and Compression

A growing interest has been developed in recent years to apply DRL techniques to optimize operations of wireless network [60]. Incorporating RL and DL into MIMO detection has evolved as a promising method for future wireless communications [71].

An RL-based cognitive BF scheme for co-located MIMO radars is proposed in [72]. The RL-empowered optimization algorithm enables the MIMO radar to iteratively sense the radar scene involving an unknown number of targets where the angular positions of targets are unknown. Therefore, the protocol synthesizes a set of transmitted waveforms whose relevant beam pattern is tailored to the learning. A BF algorithm based on online model-free RL approach SARSA is introduced in [73] to study the case of multi-target detection for a Ma-MIMO cognitive radar when the disturbance is unknown. The study has shown that RL enabled method able to detect the targets in a dynamic environment with very low SNR. The work may be improved further by refining the DoA estimate of the detected targets having a disturbance with unknown distribution.

A DNN architecture is presented in [74] in the context of MIMO detection. The authors have considered both cases, i.e., constant MIMO channel and multiple varying channels. A DNN model is also used for MIMO detection in [75]. Two architectures have been proposed, i.e., “a standard fully connected multi-layer network and a Detection Network (DetNet)”. The model of DetNet is designed by unfolding the iterations of a projected gradient descent approach into a network. The authors of [76] propose a model-driven DL model for MIMO detection. The model is particularly designed by unfolding the iterative algorithm. Deep unfolding is employed in [77] for MIMO detection for QPSK and BPSK constellation cases. A new DL-based detector using BP algorithm belief propagation is discussed in [78], which combines the belief propagation method with the DL techniques. The MIMO factor graph is used for the detection of signals from the transmitter by pressing likelihood ratios messages.

A quasi-static flat channel with many antennas is considered for detection of multilevel modulation symbols using DNN in [79] and partial learning using NN detection method for Ma-MIMO is given in [80]. A CSI sensing and recovery framework is proposed in [81]. The method learns to use channel structure effectively from training samples and authors have shown through experimental results that their method can recover CSI with reasonably better reconstruction quality. This work was further extended in [82] by developing a real-time CSI feedback framework known as “CsiNet-LSTM”. CsiNet-LSTM significantly improves recovery quality and enhances the trade-off between complexity and compression ratio by learning directly spatial structures that are combined with time correlation from training samples of Ma-MIMO with time-varying channels.

The authors of [83] have studied the feasibility of using DL techniques for automatic classification of modulation types of received signals. For example, an end-to-end CNN-based automatic modulation classification is proposed in [84] that extracts features automatically from the long symbol-rate observation sequence along with the estimated SNR. A “deep complex-valued convolutional network” that does not rely explicitly on the Fourier transform is designed in [85] to recover bits from time-domain orthogonal frequency-division multiplexing signals. A novel feedback network CRNet is presented in [86] using advanced training techniques to get superior performance by extracting CSI features on multiple resolutions.

A likelihood function learning technique for MIMO systems having a one-bit analog-to-digital converter is proposed in [87] using an RL method. The main idea of the work is to use input–output samples that are obtained by detecting the data and conduct compensation in the likelihood function for any mismatch. In another work [88], a robust MLD technique is considered for an uplink Ma-MIMO communication system with low precision ADCs under non-perfect CSI at a receiver. The same problem is addressed in [89] for a wideband SIMO system with one-bit ADCs.

A letter in [90] considers the case of uplink Ma-MIMO network with 1-bit ADCs to develop a DL-based channel estimation mechanism where the prior channel estimation record and DNN are able to learn the non-trivial mapping from quantized received measurements to channels. A DL framework for channel estimation and detection has been investigated in [91], and the results suggested that the proposed DL schemes lead to better performance in the large SNR regime. However, the architecture provides good results only for relatively small MIMO dimensions. Another detection method for MIMO optimized by back-propagation NN is given in [92] to uplink the Ma-MIMO systems.

Two DNN based detectors, i.e., damped belief propagation (BP) and max-sum by unfolding damped BP [93] and max-sum BP [94] methods, respectively, are designed in [95]. However, the framework may be further improved for optimization of the DNN architecture and efficient training schemes. Joint signal detection and channel estimation of a MIMO system using model-driven DL architecture are performed in [96]. The network for signal detection is designed by unfolding the Orthogonal AMP detection method. Due to the requirement of a few adjustable parameters for optimization, the proposed framework can be easily and efficiently trained.

A MPD using DNN is introduced in [97] by modifying the message passing detector technique to adjust the approximation error. The modification was done to achieve good performance and accelerate the convergence. The proposed architecture is then designed by unfolding the modified MPD, and the DL method is used for the optimization of the modification factors. The scaling of DNN-enabled MIMO detectors [98] in terms of systematic complexity was performed in [99]. The framework applies a part of the DNN inputs by scaling their values via weights that follow monotonically non-increasing functions. The architecture is further improved by employing a sparsity-inducing regularization constraint along with trainable weight-scaling functions. This improvement enables the model to keep a balance between detection accuracy and complexity and at the same time, robustness to variation in the activation patterns increases.

The performance of a large-scale MIMO receiver is investigated in [100]. The deployment of the MIMO receiver is done using DNN and a low-density parity check code to detect and decode disturbed signals. The model experimented with different large scale MIMO configurations to get a trade-off between the performance and complexity. An RL-based detection method for time-varying MIMO systems having one-bit ADCs is developed in [101]. Input–output samples that are being received from data detection are exploited to perform tracking of the temporal variations of likelihood functions. An MDP is modeled to handle the uncertainty of the information due to a data detection error and it enhances the accuracy of the likelihood function. In the end, an RL algorithm is used to solve the modeled MDP with less computational complexity.

Implementation methodologies for conventional MIMO transmitters of DL-based signal detection are presented in [102]. The authors have used a DNN architecture of a one-tap MIMO channel for signal detection while CNN and RNN models are applied with a multipath fading channel. A DL framework for the detection of MIMO signal for high-speed railway case is given in [103]. The proposed architecture is divided into two steps: offline training process and online detection. Another detection scheme for large-scale overloaded MIMO systems by employing DL is given in [104], where the optimization of trainable internal parameters can be performed by using standard DL schemes, i.e., SGD and back-propagation methods.

Two scenarios of channel information at the receiver using the DL model are considered in [105] for uplink pilot-assisted MU-MIMO systems. In the first case, the channel matrix is available at BS and the DNN is used as a detector and the channel matrix in the second case is not known at BS. Signal detection for Ma-MIMO systems is performed in [106] using a DL-based trainable AMP scheme. The trainable AMP model includes a preprocessing layer and a few detection layers. Moreover, the network adds trainable parameters to control prior mean-variance of MMSE denoiser. Another method for MIMO detection known as MMNet is designed in [107]. MMNet’s architecture is based on the idea of iterative soft-thresholding methods and applies a new training scheme that takes advantage of spectral and temporal correlation in real channels to speed up the training.

The BP-MMSE-based algorithm is proposed in [108] that initiates from the MMSE solution and updates the prior in every iteration with the loopy BP belief. The graph NNs are used to prevent the complexity of computing MMSE, we use Graph Neural Networks (GNNs). The graph NN model learns a message-passing scheme to solve the inference problem on the same graph. A simplification with three improvements is introduced in [109] to simplify the detection network. The first improvement is the reduction of the number of inputs, and the second is changing the network from full to sparse connectivity and decreasing the number of network layers by 50% to simplify network connection structure. While the final improvement is to optimize the loss function to prevent irreversible issues with the matrix. With these improvements, the network complexity may be reduced from $O(64n^{2})$ to O(3n).

A DL empowered detector is designed in [110] that after an off-line training able to detect signals communicated in a channel with impulsive noise. The proposed detector has less complexity than the average sphere decoder complexity and shows better performance. A Multisegment mapping model based on DNN for Ma-MIMO detection is proposed in [111]. The proposed network is developed by optimizing the prior detection networks (ScNet and DetNet) and minimizes network complexity. The authors have also introduced an activation function to enhance the performance of Ma-MIMO detection for high-order modulation cases. Tabu search detection in Ma-MIMO systems is considered in [112]. First, A DNN model for symbol detection is proposed by optimizing the DetNet and ScNet networks. Then, a tabu search algorithm based on DL is presented, where the starting solution is approximated by the first DNN model.

A likelihood ascent search detection approach based on CNN is given in [113] by using a graphical detection type for uplink multiuser Ma-MIMO systems. The proposed method needs lower average received SNRs to achieve better BER performance. Signal detection in MIMO-OFDM system has been addressed in [114] by applying extreme learning machine and autoencoder network. This combined architecture does not require the channel matrix for the signal detection process. A novel NN architecture, radial basis function networks are proposed in [115]. The model is optimized by a quantum genetic algorithm and is used to solve signal detection issues in MIMO-OFDM.

### 4.2. Channel Estimation

RL techniques, in particular, the DL method, have proven important tools to improve the accuracy of channel estimation to enhance the performance of Ma-MIMO [116] utilized in different applications such as Virtual reality, 5G systems, Internet of Things, and autonomous driving [117]. These applications demand the design of wireless systems [118] to provide ultra-low latency, large numbers of connections, and ultra-high data rates [119]. MIMO systems especially large-scale MIMO are one of the potential candidates to meet these requirements. This subsection will present the application of RL and DL for MIMO channel estimation.

Direction-of-arrival and channel estimation using RL techniques for Ma-MIMO systems are discussed in [120]. Authors have employed a DNN to realize end-to-end performance and conduct an online–offline learning mechanism. This is an efficient way of learning wireless channel statistics and the spatial structures in the angle domain, while only direction-of-arrival estimation using DL was considered in [121]. A DL method is used in [122] for channel estimation in ultra Ma-MIMO systems. The methodology can address the issues of high processing complexities and computation time with estimation efficiency and accuracy.

Channel estimation for double directional channels with limited feedback for mmWave Ma-MIMO is done in [123]. The BS estimates the downlink channel by recovering a low-rank matrix, using samples of the compressed channel matrix and feedback from the mobiles. This results in the prevention of doing resource-consuming tasks for users. The letter in [124] applied DL for estimation of the uplink channels for mixed ADCs Ma-MIMO systems. Authors have considered some antennas equipped with high-resolution ADCs and some use low-resolution ADCs at the BS. First, a direct-input DNN is used to estimate channels by employing the received signals of all antennas. Then, a selective-input prediction DNN is applied for elimination of the adverse impact of the coarsely quantized signals. Similar work was also done in [125] where low-resolution, ADC-quantized received pilot signals are used with one of three different methods: (1) “High-resolution quantized pilot + All low-resolution quantized pilot (High + All)”, (2) “High-resolution quantized pilot + Argument of low-resolution quantized pilot (High + Arg)”, and (3) “High-resolution quantized pilot + Modulus of low-resolution quantized pilot (High + Mod)”.

Channel estimation with few-bit ADCs in MIMO systems is the focus of the work in [126]. A DNN is considered first and then trained as a nonlinear MMSE channel estimator. Next, the authors used a DNN for concurrent optimization of the MMSE channel estimator and the training signal. The work in [127] also considers MIMO systems with low-resolution ADCs and proposes DL based channel estimation method. Several deep multimodal learning-enabled frameworks in Ma-MIMO systems for channel prediction are developed in [128] by taking advantage of fusion levels and modality combinations. This work on channel prediction in massive MIMO provides a significant example that may be followed in designing different deep multimodal learning-based communication approaches.

A new concept of channel mapping in space and frequency is introduced in [129]. The mapping has been done between (1). Channels at one group of antennas and one frequency band and (2) channels at another group of antennas and frequency band. Authors have first proved the existence of such channel-to-channel mapping under certain conditions and then use DNN for learning this non-trivial channel mapping function efficiently. The joint impact of nonlinear hardware impairments at UEs and the BS on the uplink performance of single-cell Ma-MIMO in practical Rician fading channel is considered in [130] using DL approach. The quality of estimation is improved as compared to distortion-aware and distortion-unaware Bayesian linear MMSE.

A new channel estimation framework is proposed in [131] with the assistance of DL technology to improve the channel estimation that is received by the least-squares method. Authors have used a MIMO system with a multi-path channel case for simulations in 5G-and-beyond networks for the scenario of mobility expressed by the Doppler effects. Although, the architecture is developed for an arbitrary number of transmitter and receiver antennas, but it can be generalized. In another work [132], the potential application of NN to optimize a particular physical layer block in a communication system by considering some salient characteristics of the emerging radio methods based on 5G standards such as beamforming, MIMO, and mmWave are investigated.

A DL-enabled method for channel estimation to improve the performance of the Ma-MIMO system is given in [133]. The used approach improves the recovery quality and enhances the trade-off between the complexity and compression ratio of the Ma-MIMO system. The method based upon using the CSI network combined with the gated recurrent unit and dropout technique scheme was used to minimize overfitting during the learning process. However, the use of the gated recurrent unit layers can increase the complexity due to the subsequent expected increase in the run time. The authors of [134] propose two channel estimation schemes using DL in TDD Ma-MIMO systems under the presence of pilot contamination and claim to have better performance than the least-squares and the covariance estimation methods in terms of the channel normalized mean square error for imperfect timing synchronization and perfect one.

The theoretical analysis on the use of DL in MIMO system channel estimation is discussed in [135]. Authors have first interpreted DL-based channel estimation by considering multiple antennas system under linear, nonlinear, and inaccurate channel statistics and have shown that DL estimator equipped with a rectified linear unit DNN is equal to that of a piecewise linear function mathematically. The Bayesian learning method is used for the estimation of channel parameters only of the interesting links in the desired cell only for the interference connections from adjacent cells [136]. The authors have claimed the possibility of obtaining an accurate estimation of the channel parameters with the exploitation of the propagation properties of Ma-MIMO systems.

Channel estimation when both low-resolution ADCs and hybrid analog–digital processing in Ma-MIMO systems are utilized is considered in [137]. The authors have formulated the quantized sparse channel estimation into a sparse Bayesian learning architecture and provide the solution with variational Bayesian technique. A Bayesian channel estimation method based on the AMP algorithm in Ma-MIMO systems is introduced in [138], and this framework requires statistical CSI. The derivation of the covariance is done for CSI acquisition by analyzing the channel model in the beam domain.

A DL-enabled architecture for channel estimation and joint pilot design of MU-MIMO channels is given in [139]. The pilot is designed using two-layer NNs and the channel estimator is modeled using DNNs and they reduce MSE of channel estimation after join training. Authors have also applied successive interference cancellations to minimize the interference that may be present among the multiple users. The pilot design for the MU-MIMO system using DL technology is also considered in [140] to reduce the sum of MSE of channel estimation where the pilot signal of every user and the power assigned to every pilot sequence is represented as a weighted superposition of orthonormal pilot sequence basis and corresponding weight respectively. Moreover, DNN is used for the optimization of power allocation to every pilot pattern to reduce the sum MSE where the input is channel large-scale fading coefficients and the pilot power allocation vector is the output.

A decision-directed method for channel estimation using DNN is developed in [141] for the MIMO system. The work was done for STBC MIMO systems by considering the highly dynamic vehicular scenario. DNN was used for k-step channel prediction for STBC while the DL empowered decision-directed channel estimator removes the requirement for Doppler rate estimation where channels are time-varying quasi-stationary. Hybrid precoding and channel estimation for multi-user mmWave MIMO system are considered in [142] using DL compressed sensing method. The prediction of beamspace channel amplitude is performed by training offline the channel estimation NN using simulated environments and the reconstruction of the channel is done using the acquired indices of entries of the dominant beamspace channel. Then, after the channel estimation, the quantized phase hybrid precoder is developed using DL and its training is performed offline with approximate phase quantization.

Channel estimation with received SNR feedback is investigated in [143] to estimate the MIMO channel coefficients using the received SNR feedback from a receiver at a transmitter based to reduce the MSE. Authors have considered time-varying fading and quasi-static block fading cases for their experiments. In another study [144], fast and flexible denoising CNN has been used for channel estimation. The framework is suitable for a wide variety of SNR levels with a variable noise level map in the shape of input. Channel estimation for Terahertz Ultra-Ma-MIMO Systems with array-of-subarrays is considered in [145] using a fifteen-layer deep CNN spherical-wave scheme. The training labels are phase shift matrix, the amplitude of the channel gain, and spherical-wave channel parameters including elevation and azimuth angles. In the end, supervised learning is employed with a self-defined loss function using the labeled data and the training of the deep CNN is performed offline and implemented online to conduct channel estimation.

Channel estimation using DL for MIMO systems for the case of multi-cell and interference-limited is taken into account in [146]. The MIMO system considered in this study is equipped with BSs and each BS serves many single antenna UE with a large number of antennas. DNN is used on the deep image prior system to denoise the received signal first and conventional least-squares estimation is done. A blind wireless channel and bandwidth-efficient estimator for the uncoded space-time labeling diversity system is designed in [147]. An NN-ML channel estimator with transmitting power-sharing is used to perform blind channel estimation for the given system and to reduce the bandwidth utilization. A wideband low-complexity DOA estimation scheme for Ma-MIMO systems is given in [148] using principal component analysis NN. The framework prevents complex angle pre-estimation by designing a focusing matrix and minimizes the complexity of the eigenvalue decomposition by using the signal subspace estimation method. Another DL-based channel estimator is designed in [149] by considering the impact of hardware impairments in a multiple-antenna BS and UEs on the uplink performance.

Effective interference cancellation and reliable channel estimation are necessary for improving the performance of MIMO UAC systems. A single carrier MIMO UAC has been considered in [150] to study a robust receiver framework using Bayesian learning for iterative channel estimation that is embedded in Turbo equalization. The proposed architecture updates the joint estimates of a channel covariance matrix, residual noise, and channel impulse response. Authors have also designed a “low complexity space-time soft decision feedback equalizer” using Bayesian learning with successive soft interference cancellations. Bayesian learning is also used in [151] as sparse learning via iterative minimization for the MIMO UAC system. The authors have also implemented a linear MMSE enabled symbol detection method by using conjugate gradient scheme and diagonalization characteristics of circulant matrices.

A MIMO receiver with inadequate pilots in a fast fading channel is considered in [152] as well as a DL-empowered Turbo-MIMO receiver that contains channel decoding, signal detection, and modules. A short pilot sequence is used to generate an estimate of the channel matrix by applying the linear MMSE method. Then, a re-estimate is done with the support of symbols that are reliably estimated. Data symbols are re-detected using the channel decoder’s soft statistics. Signal detection is performed at the receiver by applying the expectation propagation technique as multi-layer deep feed-forward networks. A blind channel estimation scheme based on DL technology is designed in [153] for OFDM-based large-scale MIMO systems. A denoising CNN has been deployed to mitigate the remaining channel and noise effect. This will help to detect the transmitted data symbols accurately at the channel sounding step. The CSI of all used was then detected as virtual pilots at each BS antennas.

A task-oriented quantization model with scalar ADCs based on DL for MIMO channel estimation is designed in [154]. The proposed quantization system does not require explicit recovery of the system model and proper quantization rule. Two receiver designs—pilot-assisted and model-drive—using DL for uplink MIMO systems are proposed in [155]. The formal receiver is developed using a data-driven full connected NNs and the transmitted signal can be recovered directly in this scheme in an end-to-end manner without the need for channel estimation. The later receiver combines communication knowledge with DL and divides the MIMO receiver into signal detection subnet and channel estimation subnet. Moreover, the application of CNN estimators has been studied in [156] for MIMO-OFDM channel estimation. The channel values of the reference signal are interpolated to estimate the channel of the full OFDM resource element matrix. Authors have developed a 2D CNN model using U-net, and a 3D CNN structure to tackle spatial correlation.

A joint design of channel estimator and pilot signal based on data-driven DL methodology for wideband Ma-MIMO systems are designed in [157]. High-dimensional channels are reconstructed using DL from underdetermined measurements by exploiting the MIMO channel’s angular-domain compressibility. The DNN architecture is employed for downlink multiuser precoding, feedback, quantization, and distributed channel estimation for a FDD Ma-MIMO system in [158], where a BS serves many mobile users. However, the feedback from the users to the BS is rate-limited. A feedback and channel estimation mechanism is developed in [159] using DL. The framework can estimate, compress and reconstruct downlink channels for FDD Ma-MIMO systems.

### 4.3. Positioning, Sensing, and Localization

MIMO systems are meeting the growing need for reliable and faster communications in wireless systems having a large number of terminals. MIMO systems can also be used to estimate the position of a terminal utilizing multipath propagation in multiple antennas. CSI-based user positioning systems using DL technology achieving a high accuracy without any overhead have shown great potential in the MIMO system. These systems are capable of positioning indoor users in both LoS and non-LoS environments with reasonable accuracy. Moreover, with the availability of smart devices and recent development in AI-based wireless systems [160], localization is enabling many location-based applications [161,162]. Demand for localization has increased because of more application of high accuracy, high bandwidth, and location-based services [163]. These applications need precise localization of users to enhance BF and resource management [164]. In this subsection, we will review DL-based architectures for exploiting MIMO-based CSI to improve indoor localization and positioning.

Fingerprinting has been an emerging research area for indoor positioning due to its location-related characteristics and ubiquity [165]. A novel DL method for Ma-MIMO fingerprint-based positioning has been investigated in [166]. They have used CNNs with a feedforward structure and measured channel snapshots. CNNs can compactly summarize and generalize relevant positioning information for channels with large data sets, e.g., in highly clustered propagation cases with Line of Sight (LOS) or without LoS. Similarly, an “angle-delay channel amplitude matrix” method for extraction of fingerprint and a deep convolutional NN-based localization scheme for Ma-MIMO-OFDM systems are proposed in [167,168]. The latter method can overcome the error of modeling for fingerprint similarity calculation.

A deterministic “uplink-to-downlink mapping function” is revealed in [169] and a sparse complex-valued neural network is proposed for the approximation of this function, where the position-to-channel mapping is bijective. The authors have demonstrated that the proposed architecture performs well as compared to the other network in terms of prediction accuracy with remarkable robustness. Later on, some of these authors have formulated the downlink channel prediction as a deep transfer learning problem and introduced the direct transfer method [170] using the fully-connected NN framework, when the network is trained in the form of classical DL and fine-tuned for new environments afterward.

The achievable rate of the mmWave Ma-MIMO system can be improved by minimizing training pilots using the DL model for sensing joint channel and hybrid precoding framework [171]. According to this study, the channel encoder first considers the NN to improve the channel sensation vectors to strengthen the sensing ability on its successful attempts, and then the precoder predicts the hybrid framework RF BF/combining vectors directly from the received sensing vector. Similar work on joint channel sensing and hybrid BF for mmWave Ma-MIMO systems using DL techniques is considered in [172]. The encoder learns to optimize the channel sensing vectors to concentrate the sensing power on the potential directions, while the precoder learns to predict the RF BF/combining vectors of the hybrid framework from the obtained sensing vector directly.

A channel sounder framework that can measure multi-subcarrier and multiantenna CSI different propagation, antenna geometries, and frequency bands are introduced in [173]. The architecture can acquire a superior accuracy of more than 75 cm for LoS and is comparable to the conventional positioning schemes and achieve the same precision for the challenging scenario of non-LoS. The feasibility of an indoor positioning framework based on NNs and CSI of a large-scale MIMO system is investigated in [174]. The proposed tailored NN architecture has a feature extractor in the shape of an additional phase branch that minimized the number of trainable parameters, resulting in a reduction in the amount of target training data. The measurements were performed for indoor environments covering a big area of 80 m${}^{2}$ with up to 64 antennas.

MIMO user positioning using DL techniques that are based on only the OFDM complex channel coefficients is examined in [175]. The proposed architecture is employed on the top of available OFDM-MIMO system and does not need any extra piloting overhead. Training of the model is done in two phases: in the first step, training on simulated LoS data, and then in the second phase, fine-tuning on measured NLoS positions. This results in minimization of the necessary measured training locations and consequently decreases the attempt for data acquisition. CNN, multilayer perceptron NN and K-nearest neighbors techniques are used in [176] for localization of a MIMO transmitter in indoor–outdoor scenarios. The proposed work has won the first position among eight teams worldwide in the indoor positioning competition organized by “IEEE’s Communication Theory Workshop” by achieving an MSE of 2.3 cm${}^{2}$.

Accuracy of localization using CSI is improved by combing multi-layer perceptron NN and K-nearest neighbors techniques in [177]. Both schemes then tested for generalization aspect in different scenarios by dividing the training and validation data in a sense that the intersection is minimized as compared to the uniform random splitting. In another work of [178], deep NN was used in the development of a robust and accurate localization scheme for a distributed massive MIMO system. CSI-based positioning is discussed in [179] using CNNs a black box and experimented on the opening of the black box using. The authors have also discussed the advantages and disadvantages of the use of an open dataset collected in a real scenario 64-antenna Ma-MIMO system. The position of a user using the CSI is inferred through CNN and then evaluated on a dataset that consists of indoor Ma-MIMO CSI measurements of three different antenna configurations, i.e., covering a (2.5 m × 2.5 m) indoor area [180]. The CNN model can be trained for the estimation of the user position inside (2.5 m × 2.5 m) with an average error of less than half a wavelength.

Indoor localization based on DL and CSI for 28 MIMO antenna is considered in [181]. The input to the multi-layer perceptron NN is the change in the magnitude component of the CSI and the learning process is improved using data augmentation. To enhance the estimation of the position, an ensemble NN scheme is applied to process the predictions of the MLPs. An improvement in indoor positioning is done in [182] by exploiting the MIMO-CSI using the proposed CNN architecture. The performance of the proposed three CNN variants is then compared with five state-of-the-art NN schemes in terms of accurate estimation of position. User positioning in OFDM Ma-MIMO systems based on 3D CNN is considered in [183,184] when the BS has a uniform planar array and traditional fingerprint type is replaced with the “angle-delay channel power matrix”. This methodology is beneficial to positioning as it embeds angles, with power in the horizontal and vertical directions.

Dynamic localization using predictive RNN for Ma-MIMO systems is investigated in [185]. The authors have designed dynamic localization structures in time-varying environments to perform localization and demonstrated the performance of the proposed architecture in indoor and outdoor scenarios with reasonable localization accuracy. Accurate mmWave positioning is important, and recently few works have been done in this area using DL techniques [186]. Positioning in mmWave Ma-MIMO using DL is considered in [187], and different NN models are applied over beamformed fingerprints such as CNN, Deep Convolution GP, LSTM, and GP LSTM to reduce the location error in the outdoor environment near to 1 m. An actor–critic RL scheme is proposed in [188] using NN approximator affine MIMO nonlinear discrete-time type systems, where disturbances and functions are not known. One NN is used as an action network to produce the best control signal while the second NN is used as a critic network cost function approximation.

### 4.4. CSI Acquisition and Feedback, Security, and Robustness

MIMO communication systems are a major enabler of the excessive throughput requirements NGN, e.g., 5G due to its ability to serve many users at a time with high energy and spectral efficiency. However, the MIMO system requires timely and accurate CSI, which is obtained by a training process including the transmission of a pilot, estimation of CSI, and feedback [189]. The training process experiences a training overhead, that scales with variation in the number of subcarriers, users, and antennas. Therefore, minimizing the training overhead in MIMO systems has been an important area of research over the last few years. Recently, DL-enabled methods have been used to reduce the overhead in feedback and CSI acquisition and have shown significant improvement compared to traditional schemes [190,191]. Here, we will present state-of-the-art DL frameworks used for CSI acquisition and feedback. This subsection will also review literature work on MIMO security and robustness aspects.

A DL approach for secure MIMO communications has been employed in [192] by exploiting the advantage of CNN learning network to generate more accurate CSI and to reduce the BER of the receiver. Both the ideal CSI and imperfect CSI are included in the training set that then may be used in different scenarios. An RNN-based DL approach is proposed in [193] to learn temporal correlation. The architecture uses depthwise separable convolution to shrink the network. Results have shown a reasonable performance in terms of recovery accuracy and quality and obtain considerable robustness at low CR.

Time-varying features have been exploited in [194] using two modules: recurrent compression/uncompression to provide an approach to minimize the parameter size. The work is extended to MU-MIMO by separately assigning a decoder network for every user at the BS. A DL-based scheme for channel calibration in Ma-MIMO systems in nonlinear settings is proposed in [195]. The framework is able to exhibit robustness in generic nonlinear scenarios even with the limited number of training sequences.

A CSI feedback mechanism based on bi-directional reciprocal channel properties and limited feedback is introduced in [196]. The Ma-MIMO BS uses the uplink CSI to recover the unknown downlink CSI from low-rate UE feedback. The DL enables architecture to minimize the CSI feedback payload significantly based on the multipath reciprocity. A DL-based denoise network is designed in [197] to enhance the channel feedback performance, and it has shown good performance at low SNR.

A CS and DL empowered CSI feedback framework for FDD Ma-MIMO communication system is propose in [198] where the CSI is compressed first at the UE using on CS scheme and then at BS CSI is reconstructed using a DL enabled signal recovery solver. Deep autoencoder is used in [199] to study CSI feedback in the FDD-MIMO system by considering the feedback delay and errors. The autoencoder is constructed by using the CSI feedback process, which contains feedback transmission delays and errors. The proposed architecture claims to minimize the effect of the feedback delay and errors.

The work in [200] presents a multiple-rate CS-NN model for compression and quantization of the CSI. The authors have adopted two network design principles and develop a novel quantization mechanism and training scheme. The proposed model improves the network feasibility by reducing the storage space at UE and enhancing the reconstruction accuracy. Similarly, a quantization method and training framework for CSI feedback using DL are given in [201]. The model uses the current CSI feedback in a real communication network and but reduces the introduced quantization distortion to enhance the quality of reconstruction.

A Bayesian CS-based feedback scheme for MIMO systems is considered in [202]. The wireless channel used in the study is time-varying temporally and spatially correlated vector autoregression. The relationship between downlink capacity and the feedback rate is obtained in closed form in statistics to perform rate-adaptive feedback. A CNN-enabled analog feedback method that maps the downlink CSI to uplink channel input directly is given in [203,204]. The DL channel estimate is reconstructed by another CNN-based corresponding noisy channel output. The framework gives a low-latency solution for rapidly changing MIMO channels because the model does not need explicit modulation, coding, and quantization.

A compression technique for channel state matrix using DL that is consists of convolutional layers and quantization and entropy coding blocks come after is proposed in [205]. The model enhances the quality of CSI reconstruction even at significantly low feedback rates. The distributed version of this work for an MU-MIMO environment is proposed in [206], where each user compresses its CSI matrices in a distributed form and reconstruction is done jointly at the BS. The Distributed version not only uses the inherent CSI pattern of a single MIMO user, but also supports the channel correlations among neighbor MIMO users.

CSI reporting which is important for MIMO system transceivers to acquire energy efficiency and high capacity in FDD form is considered in [207] using DL technology. The proposed DL-based compression architecture jointly handles recovery, codeword quantization, and CSI compression under the bandwidth constraint to enhance the encoding performance of CSI feedback. The correlation between nearby UE has been exploited in [208] by designing a DL-based CSI feedback and cooperative recovery mechanism to minimize the overhead of the feedback. Authors have also proposed a baseline NN framework with LSTM for a UE equipped with multiple antennas to extract the correlation of surrounding antennas and two magnitude-dependent phase feedback schemes that present instant CSI and statistical magnitude information.

Spatial correlation-based CSI compression feedback for FDD Ma-MIMO systems is considered in [209] and a DL-based CSI compression feedback scheme is used in single-user as well as multi-user environments. The framework takes into account the spatial correlation of Ma-MIMO system uniform linear antenna arrays and takes advantage of full of the channel information during the training. Single-cell and multi-cell scenarios are also discussed in [210] in terms of CSI feedback for BF to optimize the BF performance gain instead of the feedback accuracy. The encoder at the user in a single-cell does compression of the CSI and the BF vector is generated at the decoder, while in the multicell system, two kinds of CSI feedback has to be sent, i.e., the targeted and the interfering CSI. A binarization assisted feedback NN is proposed in [211] to improve the performance under customized training and inference approaches.

Implicit feedback framework based on DL is applied in [212] to inherit the low-overhead features. The given architecture uses NNs to interchange the precoding matrix indicator encoding and decoding components. Moreover, a correlation between sub-bands is also employed for more improvement in feedback efficiency. A compressive sampled CSI feedback scheme for Ma-MIMO system using DL is proposed in [213], where the channel matrix is sampled in frequency/time dimension uniformly before feeding into NNs. This will minimize the computational time/resource at UE and improve the accuracy of the CSI recovery at the BS. A CSI feedback network using DL for the FDD-MIMO system is studied in [214], but its application to the mobile terminal is not effective due to the large numbers of parameters. Thus using the developed network, authors have designed a new lightweight CSI feedback framework. Similar work on the development of lightweight NN for MIMO CSI feedback was also done in [215]. An FDD Ma-MIMO communication system that prevents signaling overhead by applying a DL-enabled channel extrapolation is demonstrated in [216].

A scheme to protect the DL-based CSI feedback process from white-box adversarial is presented in [217]. The authors have also shown that jamming attacks may be crafted with some precautions.

A CNN-based network known as aggregated channel reconstruction model is constructed in [218] to enhance feedback performance with parametric ReLU activation and network aggregation. In particular, the elastic feedback method is introduced to flexibly adjust the network to address various resource limitations. Moreover, the network binarization scheme is integrated with the feature quantization for practical deployment. Long-range dependencies are captured efficiently in [219] by using DL-based CSI feedback method and taking benefits of non-local blocks. Additionally, the feature of the refinement part is strengthened by adopting dense connectivity. AnciNet, a DNN empowered framework is designed in [220] to manage CSI feedback with limited bandwidth. The proposed architecture extracts noise-free patterns from the noisy samples of CSI to obtain CSI compression for the feedback effectively.

An uplink-assisted downlink channel acquisition architecture using DL is presented in [221] to minimize feedback and high training overheads. The proposed framework takes into account the full downlink CSI acquisition process such as channel estimation, downlink pilot design, and feedback. A CNN model is developed on the Markovian model in [222] to encode forward CSI differentially in time to enhance reconstruction accuracy effectively. Authors have also explored convolutional layers for the compression of feedback and spherical normalization of input data. A fully convolutional NN is presented in [223] for the compressing and decompressing the downlink CSI. The proposed model enhance the reconstruction accuracy of downlink CSI and minimize the training parameters and computational parameters.

Superimposed coding and DL techniques are combined in [224] for CSI feedback. The proposed methodology first spread downlink CSI and then superimposed it on uplink user data patterns toward the BS. A NN framework is then designed for BS for the recovery of downlink CSI and uplink user data sequences. A deep transfer learning scheme is developed in [225] to addressed the high training cost of the NN used for 5G MIMO downlink CSI feedback. The proposed architecture uses a comparatively less number of samples for fine-tuning of a pre-trained model and provides the possibility to achieve a new model with reduced training cost.

A DL-based scheme for the prediction of downlink CSI in Ma-MIMO FDD systems is presented in [226]. The given architecture utilizes a complex-valued NN in a complex domain to tackle complex CSI matrices and adjusts 3D convolution operations for the extraction of features. Similarly, a model-driven DL-enabled downlink channel reconstruction design is proposed in [227] for FDD massive MIMO systems.

### 4.5. mmWave Communications

Deep learning techniques have been utilized recently for interesting and important applications in mmWave and Ma-MIMO systems. DL provides solutions to hard optimization problems due to its powerful capabilities of learning unknown models.

A DL-based compressed sensing channel estimation and quantized phase hybrid precoder design scheme is proposed in [142] for the MU mmWave Ma-MIMO communication systems. The proposed work claims to have better performance in terms of spectral efficiency as compared to other techniques having low phase shifter resolution. A novel DL-based method is developed in [228] to estimate the channel for beamspace mmWave Ma-MIMO communication systems. This framework uses a learned denoising-based approximate message passing network by taking advantage of iterative signal recovery methods and DL techniques. A DL-based analog and digital beamforming method for mmWave point-to-point Ma-MIMO system are given in [229,230] for reduction of system bit error rate and improvement in the spectral efficiency.

An integration of ML and coordinated BF scheme is employed in [231] to enable highly mobile applications for large antenna array mmWave MIMO systems. Authors have taken the benefit of DL that learns the mapping between beam training results and Omni-received uplink pilots. Similarly, another RL-based solution for mmWave Ma-MIMO system is proposed in [232] for effective hybrid precoding, where every choice of the precoders for achieving the optimal decoder is considered as a mapping mechanism in DNN. In particular, the hybrid precoder is chosen via DNN-based training to optimize the process of precoding process in mmWave Ma-MIMO.

A generic dataset for mmWave/Ma-MIMO channels known as the DeepMIMO dataset was introduced in [233] with two important features: (1) The construction of the DeepMIMO channels is based on accurately obtained ray-tracing data from the “Remcom Wireless InSite”. That means it captures the dependence on the locations of transmitter/receiver and environment geometry/materials and this is essential for many ML applications. (2) The DeepMIMO dataset is parameterized/generic so that one can adjust a set of channel and system parameters to tailor the generated DeepMIMO dataset for the target ML application. A ray-tracing [234] and vehicle traffic simulator are combined in [235] to produce channel realizations that represent 5G situations with mobility of both objects and transceivers. Authors have used a particular dataset to investigate beam selection method on vehicle-to-infrastructure using mmWave MIMO.

A hybrid processing model for the mmWave Ma-MIMO system is normally employed to minimize cost and complexity. However, channel estimation may be very challenging through this method. The work in [236] presents a deep CNN that can exploit both the frequency and spatial correlation, while the input into the CNN has corrupted channel matrices at adjacent subcarriers simultaneously. The same research group used Deep CNN to carry out estimation for the wideband channel of mmWave Ma-MIMO systems in [237]. The proposed scheme exploits the frequency correlation in addition to the exploitation of spatial correlation and here the input into CNN are channel matrices estimated tentatively at multiple adjacent subcarriers.

Beamforming gains are achievable and high path loss is preventable in mmWave systems by deploying a large number of antennas. However, with a large number of antennas, implementation of digital precoders is difficult due to hardware constraints and, at the same time, analog precoders have performance limitations. Hybrid precoding is an important task in mmWave MIMO systems to reduce the cost and complexity as well as to obtain a sufficient sum rate. Greedy methods or optimization techniques have been used in literature for hybrid precoding. However, these schemes depend on the channel data quality and achieve sub-optimum performance, and also give higher complexity. Therefore, in the next few paragraphs, we will discuss few proposals on the use of DL-based hybrid precoding. Some alternating algorithms for hybrid precoding for mmWave may be seen in [238].

Two schemes, i.e., CNN-based and equivalent channel precoding, are designed in [239] for mmWave Ma-MIMO systems. The complexity is decreased significantly with equivalent channel precoding but the performance is a little less than full digital precoding while the CNN precoding method shows much better robustness to imperfect CSI. In another related work [240], authors have considered the case of multi-user and proposed a DNN based hybrid BF system. They have simultaneously inferred users and sub-optimal beam codewords of the BS by applying the received signals only on the target RF beamformers and hence reducing the complexity of beam training.

A DL empowered hybrid precoding architecture is proposed in [241] that uses large-scale information for the prediction of decoder and hybrid precoders parameters. The statistics of the channel covariance matrix are applied to design the hybrid precoders and decoders. The architecture is able to optimize the hybrid precoder and decoder in terms of maximum spectral efficiency after training. Moreover, a CNN architecture for the joint design of precoder and combiners is designed in [242] that takes the input of the channel matrix and returns the output of analog and baseband beamformers. The underlying CNN scheme does not need knowledge of steering vectors of array responses and it achieves higher capacity performance.

Hybrid precoding for MU-MIMO system is done in [243] using CNN. The framework accepts an imperfect channel matrix as input and at the output gives the combiner and analog precoder. An exhaustive search algorithm is developed first, to chose combiners and the analog precoder from a predefined codebook. In the second step, combiners and precoder are employed as output labels during the training network.

Fully convolutional denoising (FCD) AMP scheme is introduced in [244] by combining FCD networks with learned AMP networks in mmWave Ma-MIMO system by considering NN framework able to learn channel patterns and extract noise features. A beamspace channel estimation method using prior-aided Gaussian mixture DNN empowered learned AMP is given in [245]. A prior-aided GA beamspace channel estimation method using prior-aided Gaussian mixture learned AMP network is designed by replacing the original shrinkage function with that of the derived Gaussian mixture for accurate estimation of the beamspace channel.

A DL-CS channel estimation method consisting of channel reconstruction and beamspace channel amplitude estimation is given in [246]. The NN is trained offline based on simulated environments using the mmWave channel model and the correlation between the measurement matrix, and the received signal vectors are applied as input to the trained NN which is used for the prediction of the beamspace channel amplitude. Then, reconstruction of the channel is done using the acquired indices of entries of the dominant beamspace channel. A mmWave OFDM-MIMO receiver is considered in [247] with a generalized hybrid structure where RF chains and low-resolution ADCs are deployed simultaneously. The authors have developed a computationally efficient Bayesian data detection scheme that gives an MMSE estimate on data symbols. The authors have also designed a low-complexity realization where only matrix-vector multiplications and fast Fourier transform are needed.

Beamforming for mmWave MIMO systems is considered in [248] using the multi-agent distributed double deep Q-learning approach, where many BSs can dynamically and automatically adaptable their beams to serve many highly mobile UEs. Authors have assumed the largest received power mapping criterion for UEs with a realistic channel model. A frequency-selective wideband mmWave network is considered in [249] with two DL compressive sensing assisted schemes. The proposed methodology learns important a priori information from training data to give the most accurate channel estimates with reduced training overhead. Estimation of a channel for mmWave MIMO system is discussed in [250]. A modified convolutional blind denoising model is developed to boost the robustness in the noisy channel by adjusting asymmetric joint loss functions, on-blind denoising subnetwork, and noise level estimation subnetwork for the blind channel estimation. Moreover, the proposed network can minimize the noise interactively by adopting the estimated noise level map in the channel matrix.

Uplink mmWave Ma-MIMO systems are considered in [251] using Bayesian learning. In this work, an angle domain off-grid channel estimation method is developed by using the spatial sparse pattern in mmWave channels. Similarly, sparse Bayesian learning is also used in [252] in hybrid mmWave systems for channel estimation. The proposed model exploits spatial sparsity in the wireless channels that exist due to a highly directional pattern of propagation. The large intelligent surface-assisted mmWave Ma-MIMO systems are the focus of work in [253] and DL architecture is proposed for channel estimation. A twin CNN framework is developed for estimating both the direct and the cascaded channels by feeding CNN with the received pilot signals. Hybrid precoding and channel estimation mmWave MIMO systems using DL has been studied in [254]. Authors have used the hierarchical codebook based method for channel estimation. An adaptable DNN based low-rank channel recovery methodology is presented in [255] for a hybrid array based massive MIMO system. The proposed framework includes a common feature extraction element and the adaptable recovery module.

### 4.6. Resource Management and Scheduling

Radio Resource management [256] and user scheduling [257,258] in MU-MIMO and Ma-MIMO is very crucial for achieving good performance and often solve using techniques from optimization theory. The heterogeneity and increased complexity of MIMO systems like Ma-MIMO demands a paradigm shift from conventional resource management mechanisms. RL and DL are powerful techniques wherein DL a multi-layer NN may be trained to model a resource allocation algorithm using available data. Therefore, there is no need for intensive online computations for resource management decisions which would be required otherwise for the solution of resource allocation problems. This subsection focuses on the applications of RL and DL on solving the problem of radio resource allocation for different types of MIMO systems.

Deep learning has been used in [259] for the prediction of the power allocation profiles for a new group of users’ positions. A DNN was used that learns the mapping between optimal power allocation policies and the positions of UE after training. A Q-learning method is proposed in [260] to maximize the overall capacity of the network when BS is densely and randomly distributed with reasonable improvement in stability and convergence speed. The authors of [261] have introduced a DL technique for “heterogeneous fifth-generation new radio networks” to improve the performance of the downlink coordinated multipoint. The proposed methodology is based on the construction of a surrogate coordinated multipoint trigger function where the cooperating set is a single-tier of sub-6 GHz heterogeneous BS operating in the FDD mode.

A universal DRL-based framework is given in [262] for access control and resource management. The framework adopts both CNN and RNN for automatically modeling the sequential features and potential spatial features from the raw wireless signal. An RL-based power control method is presented in [263] for the downlink NOMA transmission without the knowledge of radio channel parameters and jamming. They formulated the power allocation of a multiple antennas BS in a NOMA system contending with a smart jammer as a zero-sum game where the first BS selects the transmit power on multiple antennas and then the jammer chooses the jamming power to cause an interruption in the transmission of users.

A DQN-based technique is used in [264] for resource allocation in Ma-MIMO-NOMA systems. The RL method is employed for the development of an iterative optimization structure for beamforming, power allocation, and user clustering. In particular, a DQN is modeled to group the users in accordance with the reward that has been calculated after beamforming and power allocation. Moreover, as a part of the study, the use of DL for the optimization of power control in Ma-MIMO systems has been investigated in [265]. The article [266] considers deep spatial learning methods for scheduling that have the possibility to bypass the channel estimation step. The authors have applied a DNN to produce a near-optimal schedule only based on the geographic locations of the receiver and transmitters in the system.

The optimization of sum spectral efficiency for multi-cell Ma-MIMO communication systems is considered in [267] for a different number of active users. The proposed methodology employs the information of large-scale fading for the prediction of both data power and the pilot. Authors have used the problem structure to model a single NN able to handle a different number of active users that are varying dynamically. A similar optimization algorithm is presented in [268], inspired by the weighted MMSE method, to get a stationary point in polynomial time. Authors then use DL to train a CNN for performing the pilot power control and joint data in sub-millisecond runtime.

The optimization of downlink beamforming via DL techniques is for the MISO system is done in [269,270]. The method is based on CNN and exploitation of the downlink-uplink duality and the known pattern of optimal solutions. A novel DRL-based capacity and coverage optimization method is proposed in [271]. The architecture also contained a DRL-enabled user scheduling method and a novel intercell interference coordination technique to address capacity and coverage in Ma-MIMO networks. Similarly, the work in [272] discusses the use of the DL approach for load balancing and user association for sum rate optimization.

A pilot assignment technique using DL for a Ma-MIMO system equipped with a large number of antennas is given in [273] to improve the performance of cellular networks with severe pilot contamination by learning the mapping between users’ location pattern and pilot assignment. The proposed architecture was implemented through a commercially available deep multilayer perceptron model. A combination of RL and radio service maps is used in [274] to switches off BSs effectively and evaluated by utilizing a 3D ray-tracing model on computer simulations. An inter-cell interference management method for MIMO systems is presented in [275]. The framework contains interference cancellation on the receiver and NNs power control on the transmitter end. The authors have evaluated networks of MIMO systems with the power optimization using a few intercell interference coordination techniques: the belief propagation, the greedy search, and NN, combined with IC on the receiver side. Another work in [276] considers the suppression of intercell interference for OFDM-MIMO systems. The authors have employed a complex-valued NN architecture using the traditional interference rejection combining.

An intelligent algorithm to optimize the performance of the Ma-MIMO beamforming is introduced in [277]. The proposed framework combines three NNs to implement the deep adversarial RL workflow cooperatively. One NN is trained to produce realistic patterns of user mobility, being used by the second NN to generate a corresponding antenna diagram. The third NN does the estimation of the efficiency of the generated antenna diagram and returns respective reward to two networks.

### 4.7. Miscellaneous Applications

The results of some recent works indicate that deep learning models can learn a form of decoding algorithm, instead of only a classifier. These studies provide that DL architectures can be applied for improving a standard belief propagation decoder, although having large example space [278]. Moreover, identical improvements are achieved for the min-sum algorithm.

Metric normalized validation error is introduced in [279] to investigate the applications and limitations of DL-based decoding for different performance metrics, e.g., complexity. The authors of [280] present an iterative belief propagation CNN model for channel decoding under correlated noise. The framework concatenates a trained CNN with a standard belief propagation decoder. A recurrent neural decoder model using the technique of successive relaxation is introduced in [281]. The authors have observed better performance over standard belief propagation are on sparser Tanner graph representations of the codes.

A practical issue of imperfect successive interference cancellation decoding for real-world NOMA system is considered in [282]. A novel DL-based scheme is proposed by authors for the downlink of the MIMO-NOMA system where both successive interference cancellation decoding and precoding are jointly optimized. Another problem of dynamic multichannel access is discussed in [283] in which multiple correlated channels observe an unknown joint Markov model and UEs choose the channel for the transmission of data. The goal of the study is to find a policy that optimizes the aggregated future successful transmissions. The scenario is formulated as a POMDP without known system dynamics.

A novel physical layer DL scheme for MIMO system using an autoencoder is developed in [284] by using a transmitter and receiver which is an extension to the work on joint optimization of physical layer representation as well as encoding-decoding processes from to the multi-antenna case. An unsupervised DL-based autoencoder is also used in [285] for single-user MIMO communications to introduce a novel physical layer approach. The research contribution is an extension of joint optimization [286] of physical layer representation as well as the encoding-decoding processes as a single end-to-end task. The work is extended for multi-antenna cases by expanding transmitter and receivers.

A DL-based channel prediction scheme is developed in [287] to enable FDD large-scale MIMO for deployment. Authors have removed large signaling overhead using DL based channel prediction method and used a NN at the BS to infer the DL CSI that is centered around a frequency $f_{DL}$ by only observing uplink CSI on a different but nearby frequency region around $f_{UL}$. Then, there is no requirement of reporting/pilot overhead with a genuine TDD-based system. A DL-based autonomous channel measurement framework that can accurately predict channel information consisting of a few multi-path effects is developed in [288]. The architecture attains channel magnitude measurements autonomously using eight antennas through a mobile robot containing a transmitter that receives wireless commands from a central computer.

Channel characteristics are predicted in [289] using the ML method and CNN for 3D mmWave Ma-MIMO system indoor channels. Elevation angle of arrival, azimuth angle of arrival, the elevation angle of departure, azimuth angle of departure, amplitude, and delay are produced by ray-tracing software. While channel statistical characteristics can be obtained after data preprocessing to train the CNN. A DNN-enabled decoding framework for screen-camera communications and a unity 3D-based evaluation scheme is introduced in [290] to boost the obtainable throughput and to synthetically learn the DNN structure for being robust against multiple different screen-camera scenarios respectively. Jointly sparse support and jointly sparse signal recoveries have been investigated in [291] in multiple measurement vector schemes for complex signals that may appear in various applications in signal processing and communications.

An RL actor–critic enabled fault-tolerant control problem is discussed in [292] for MIMO nonlinear discrete-time communication systems. The authors have considered both abrupt faults and incipient faults. An action NN is designed to produce the optimal control signal while the cost function is approximated with the critic network. MIMO uncertain nonlinear dynamic networks having unknown varying control direction matrix and external disturbance are considered in [293] and a continuous tracking control law is introduced. The proposed framework includes a robust term, an online approximator (represented by a two-layer NN), Nussbaum gain matrix selector, and high-gain feedback.

A robust adaptive NN control is discussed in [294] for a general type of uncertain MIMO nonlinear systems having input nonlinearities and control coefficient matrices are not known. The proposed framework combines Lyapunov synthesis and backstepping with variable structure control for nonsymmetric input nonlinearities of deadzone and saturation. In another work on MIMO uncertain nonlinear systems with actuator saturation and extern disturbances [295], an adaptive controller by taking into account a priori actuator saturation effects is presented and gives the guarantee of performance tracking. Authors have used adaptive radial basis function NNs for the approximation of unknown nonlinearities. Moreover, an auxiliary system is designed for the compensation of actuator saturation effects.

ANC for uncertain MIMO nonlinear systems is introduced in [296] when input saturation and external disturbances are present. The uncertainties of the system are handled by NN approximation and unknown disturbances are tackled by the Nussbaum disturbance observer. A dynamic adaptive output feedback NN controller for MIMO affine in the control uncertain nonlinear systems is developed in [297]. The controller can guarantee prescribed performance limits on the system’s output and the boundedness of closed-loop signals.

An adaptive backstepping control approach is proposed in [298] uncertain MIMO incommensurate fractional-order nonlinear systems. Approximation of unknown nonlinear uncertainties is done by the radial basis function NN in every step of the backstepping process. An ANC scheme for MIMO nonlinear systems with different constraints is developed in this [299]. The ANC architecture is combined with disturbance observer, barrier Lyapunov function, radial basis function NN, backstepping method to tackle the constrained states, and nonsymmetric input nonlinearity. Similar works on ANC for uncertain MIMO nonlinear systems using NN are done in [300,301]. An adaptive DL empowered unmanned aerial vehicle receiver is designed in [302] for coded MIMO systems. Authors have employed the linear convolutional code at the transmitter. The proposed iterative unmanned aerial vehicle receiver consists of three parts such as ZF or MMSE detector, the deep CNN that can suppress the noise by capturing the correlation characteristics among noise, and the Viterbi decoding decoder.

## 5. Statistics and Impact

This section presents statistics about the surveyed papers and an analysis of their impact.

First, we have grouped the literature in four different periods as can be seen in Table 2. It is possible to quantify the remarkable improvement in the adoption of DL and RL over the last five years in MIMO systems. Details about the number of papers published by years, from 2010 up to 2021, are reported in Figure 3.

Next, we have considered different categories. Therefore, Figure 4 reports the total number of papers concerning different issues of MIMO communication and exploiting RL and/or DL technologies.

Figure 5 shows how the surveyed papers are distributed with respect to the analyzed categories. Note that such papers are more or less equally distributed. From Figure 4 and Figure 5, it is possible to deduce that most of the article concerns three categories (Section 4.1, Section 4.2 and Section 4.4). In these three categories, we have considered more topics and there are more papers related to them. Similarly, the number of papers related to the other three categories (Section 4.3, Section 4.5 and Section 4.6) is almost the same, but definitively lower. Moreover, there is also a significant amount of work presented in Section 4.7 that we have not listed among the previous six categories. Most of the works presented in this section are related to channel encoding–decoding and adaptive control in uncertain MIMO nonlinear systems.

Figure 6 and Figure 7 show the distribution of surveyed papers according to DL and RL architectures respectively. Figure 6 concerns DL architectures. Most of the applications rely on DNN architectures. While a significant amount of papers also take advantage of the CNN framework. We can also see the contribution of other DL architectures such as RNN, RBFNN and Autoencoder. Similarly, Figure 7 presents the exploitation of different RL schemes. Data indicates that most of the works rely on Bayesian and Q-learning.

Finally, as a further detail of data reported in Figure 8, we have also listed the top five most cited papers for each category in Table 3. This can provide information about the papers with a higher impact on the research community.

## 6. Discussion

This section discusses the surveyed papers by highlighting the strengths and limitations, along with future directions.

It is noted from results of channel estimation that DL frameworks show better results in the large SNR environments, but are outperformed by standard iterative message passing methods. Moreover, adopted DL schemes are suitable for high MIMO dimensions, but converged for comparatively small MIMO dimensions for decoding [91]. Therefore, there is the need for efficient DL architectures to handle the convergence issues that appear in time-varying channels and 1-bit quantization.

Many problems are associated with 5G Ma-MIMO technology, which can be mitigated through the use of DL. For example, it is difficult to estimate the accuracy of the channel by employing conventional estimation schemes and with a suitable number of pilots. The performance of the low complexity least-squares estimator is not satisfactory. On the other hand, MMSE channel estimation is relatively complex [70]. Therefore, DL can be applied to bypass these issues as done in [81,117,124,228,303,304,305].

In addition, within linear systems, the DL estimator is near to the linear MMSE estimator, but it outperforms this last one significantly when there is a nonlinear signal model. However, it is sensitive to the training data quality and estimation performance may degrade appreciably when the data in real regimes distribute wider than that of the training data [135]. Both the advantages and cost of the DL estimator should be considered when applying it in real wireless communication systems for channel estimation. There is a need to keep a balance between state-of-the-art channel estimation and DL-empowered channel estimation.

The computing capabilities and limited memory of wireless devices may not suite for complex DL algorithms. A considerable amount of time is required for collecting a sufficient number of samples and for training DL models. This can become a critical impediment to implement such algorithms on wireless devices with limited storage and power. Moreover, some MIMO applications need on-fly sampling as well as real-time processing, which makes training difficult. Obtaining more samples and long-time model training results in slow feedback. Therefore, DL models should be designed to acquire optimal accuracy with fewer samples and shorter periods.

User privacy is the most important concern of service providers. While deploying DL in wireless systems, one challenge is how the training is enabled on a dataset associated with users without sharing the input data and exposing personal data to risks. It is important to have a security scheme to speed up the integration of DL in MIMO communications.

Security of DL networks is another challenge, as NNs are prone to adversarial attacks. These attackers may affect the process of training by inserting fake training data, which can reduce the accuracy of the DL models. This may lead to a wrong design, which can affect the overall performance of the network. Research in the security of RL and DL techniques remains shallow.

Multiple antennas are required to mitigate high path loss and to achieve BF gains in mmWave systems. However, it is difficult to employ digital precoders in presence of many antennas due to hardware constraints. Similarly, the performance of analog precoders is limited [241]. Therefore, hybrid precoding using DL architecture is a feasible solution as the DL-based precoder takes advantage of the large-scale information for parameters prediction of hybrid precoders, as well as of decoders.

Despite the remarkable progress of DL in communication, still, research efforts are required in many directions to ease the integration of RL and RL. The acceleration of DNN alongside distributed RL systems, cloud computing, faster algorithm, and advanced parallel computing provides an opportunity for 5G to develop the intelligence in its communication systems to provide ultra-low latency and high throughput. Recently, some efforts have been done in DNN acceleration [306]. The acceleration of DNN can be at architecture, computation, and implementation levels.

Techniques such as knowledge distillation [307], projection [308], pruning [309] and layer decomposition [310] can be used at architecture level. Many characteristics may be investigated for the implementation level, including FPGA designs [311] and advanced GPU [312]. With the use of DL acceleration schemes, we can lower the complexity of DL with reduced loss in the accuracy of this architecture. A combination of these approaches may decrease the amount of parameters by more than 50%.

Furthermore, more exploration of the acceleration of these models may have a significant impact on the adoption of DL to develop intelligence in MIMO systems. Integration of DL and RL in MIMO communication systems can speed up by data collection and subsequent cleansing. With the availability of datasets, researchers can build and test their architectures. Therefore, efforts are required to build systems that can produce datasets.

## 7. Conclusions

We presented a comprehensive review of the applications of RL and DL to different issues of MIMO communication systems. First, we presented an introduction of both classes of AI methods (i.e., RL and DL) and MIMO systems. Afterward, we have presented various applications of such AI technologies in MIMO systems. Then, we have analyzed the impact of research papers in the field. Finally, we have outlined open issues, some limitations, and future research directions.

## Figures and Tables

**Figure 1 sensors-22-00309-f001:**
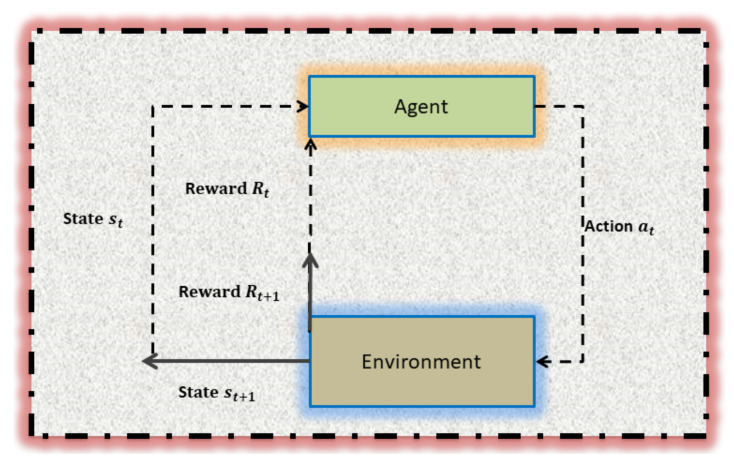
The reinforcement learning problem.

**Figure 2 sensors-22-00309-f002:**
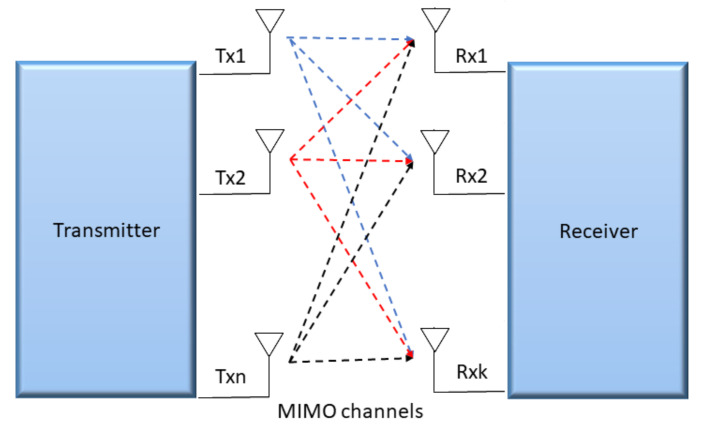
MIMO communication.

**Figure 3 sensors-22-00309-f003:**
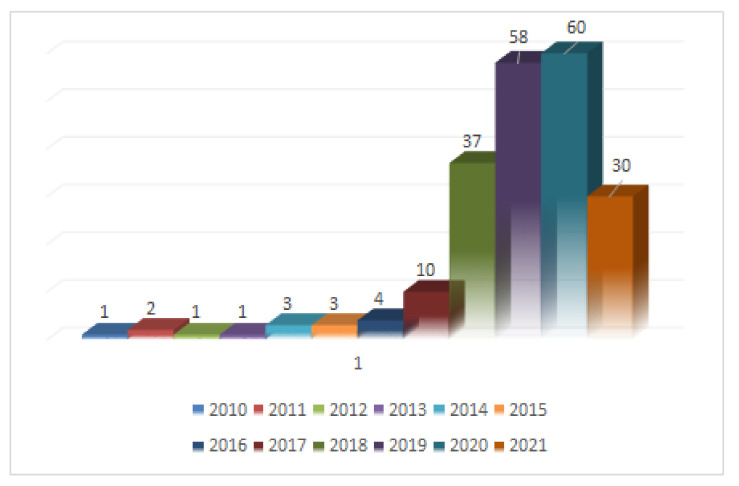
Number of papers from 2010 to 2021.

**Figure 4 sensors-22-00309-f004:**
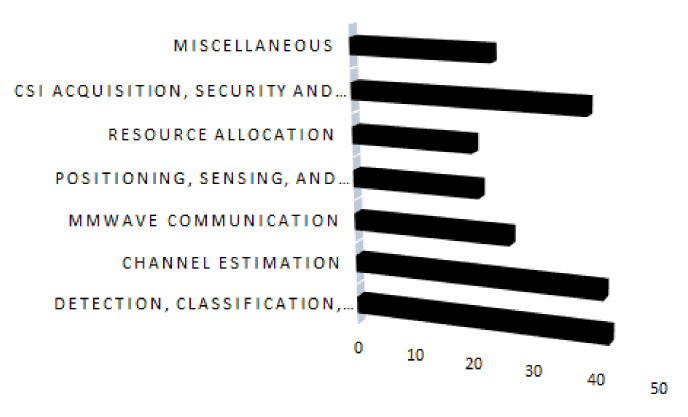
Number of papers surveyed by category.

**Figure 5 sensors-22-00309-f005:**
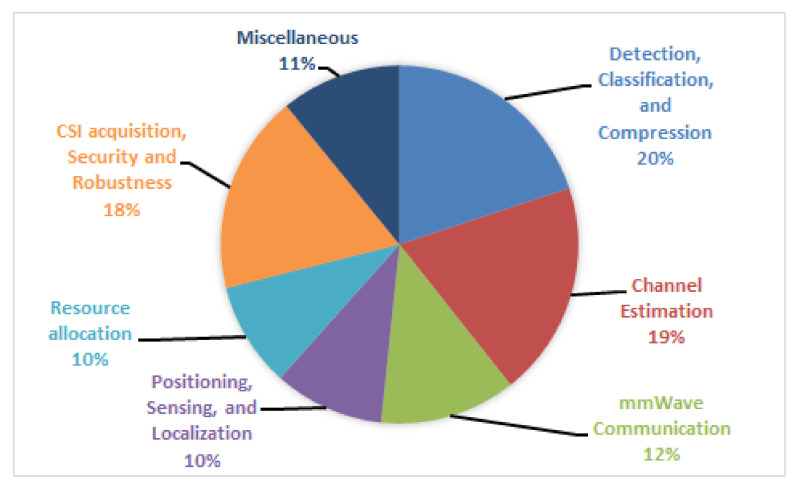
Distribution of the surveyed papers with respect to the different categories.

**Figure 6 sensors-22-00309-f006:**
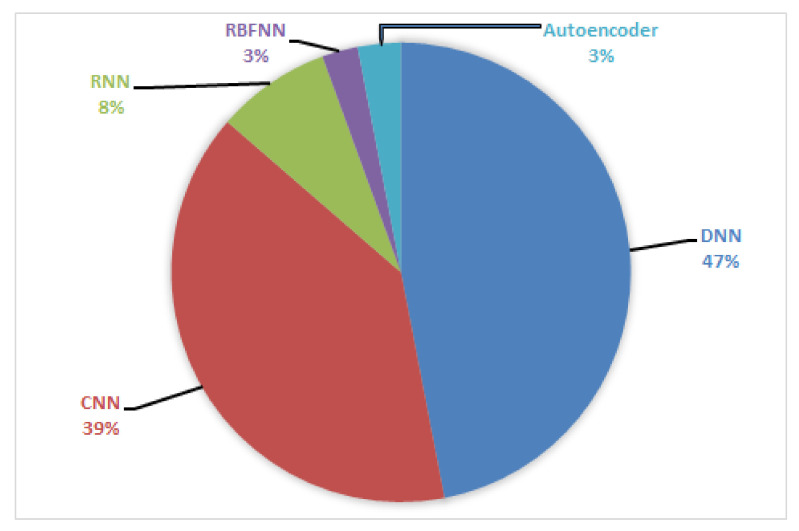
Percentage of papers surveyed by DL architecture.

**Figure 7 sensors-22-00309-f007:**
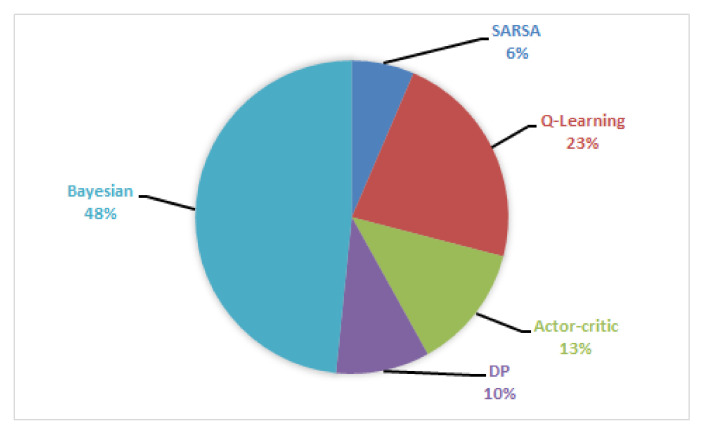
Percentage of papers surveyed by RL algorithm.

**Figure 8 sensors-22-00309-f008:**
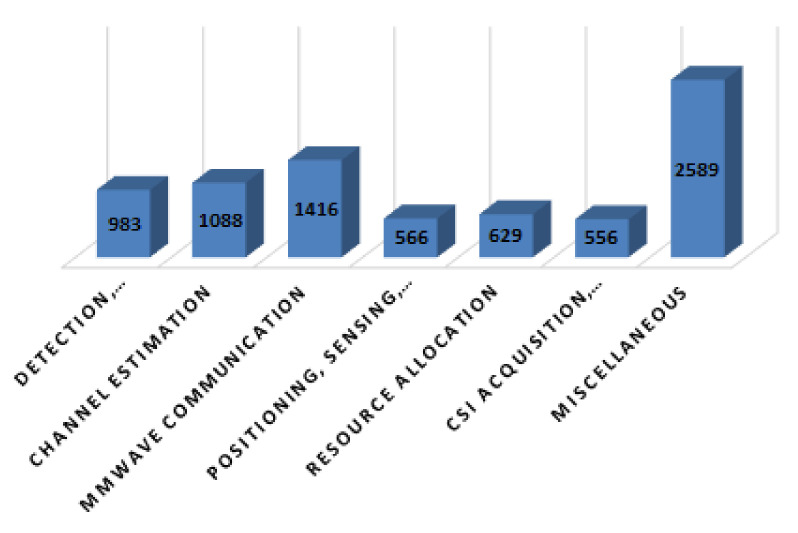
Number of citations by category.

**Table 2 sensors-22-00309-t002:** Number of papers from 2010 to 2021.

Period	Number of Papers
2010 to 2012	4
2013 to 2015	7
2016 to 2018	51
2019 to August 2021	148

**Table 3 sensors-22-00309-t003:** Top most cited papers from each category (left to right in descending order).

Category	Paper-1	Paper-2	Paper-3	Paper-4	Paper-5
Detection, Classification, and Compression	[81]	[103]	[75]	[82]	[76]
Channel Estimation	[120]	[285]	[144]	[129]	[124]
mmWave Communication	[228]	[232]	[231]	[233]	[236]
Positioning, Sensing, and Localization	[188]	[166]	[169]	[172]	[173]
Resource allocation	[266]	[262]	[268]	[273]	[277]
CSI acquisition, Security and Robustness	[200]	[194]	[196]	[199]	[193]
Miscellaneous	[294]	[279]	[281]	[136]	[283]

## Data Availability

No database has been used during this work.

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
