# Peer review of "Application of Reinforcement Learning and Deep Learning in Multiple-Input and Multiple-Output (MIMO) Systems"

_sensors, 2021, doi:10.3390/s22010309_

Round 1

Reviewer 1 Report

The comments are attached as pdf file.

Author Response

We are grateful to the Associate Editor and all the Reviewers for their valuable comments, which we have tried to integrate fully within this revised version of our paper. 

For details, please see our replies to the Reviewers’ comments.

Detailed responses to the Reviewer-1 questions are given below:

First Reviewer

Reviewer’s general comment 

In this manuscript, the application of RL and DL for the different aspects of MIMO systems has been surveyed. The potential applications of the RL and DL for different areas of the MIMO systems such as detection, classification, and compression; channel estimation; positioning, sensing, and localization; CSI acquisition and feedback, security, and robustness; mmWave communication and resource allocation are also reviewed in detail. There is a requirement for a paper on the application of RL/DL in MIMO systems. The paper is well written but not well organized

Reviewer’s question R1.1

The abstract of the paper needs revision. MIMO should be first completely defined before using its acronym. A large part of the abstract is given in the final parentheses. The parentheses material is typically perceived as supplementary.

Authors’ answer to R1.1 

 Changes have been made as advised by the reviewer.

Reviewer’s question R1.2

Move Section VI as Section III 

Authors’ answer to R1.2 

Section 6 has been revised and now moved to section 3.

Reviewer’s question R1.3

Acronyms should be in upper case, e.g., mimo in Ref [1]

Authors’ answer to R1.3 

We have revised all acronyms and used them in upper case letters.

Reviewer’s question R1.4

Some papers in the same conference have the same page number, e.g., Refs [23] and [89]

Authors’ answer to R1.4 

We have checked and found that the conference for Ref [23] was held in 2019 and conference for ref [89] was held in 2017, so both were different conferences.

Reviewer’s question R1.5

Increase font size for some figures, e.g., Figure 3.

Authors’ answer to R1.5 

We have improved the readability of all figures including Figure 3.

Reviewer’s question R1.6 

Page 4, column 2, paragraph 4, line 8: Is a reference number missing in []?

Authors’ answer to R1.6 

We have inserted the missing reference.

Reviewer’s question R1.7

Remove extra commas, e.g., page 11, paragraph 1, line 10 and paragraph 2, line 3.

Authors’ answer to R1.7 

Extra commas have been removed.

Reviewer’s question R1.8

Page 1, column 2, the last two lines: Put "Section VI" in the same line. Fix other similar instances.

Authors’ answer to R1.8 

Corrected.

Reviewer’s question R1.9

Section III.A, paragraph 2:BF is defined twice. Check other acronyms also

Authors’ answer to R1.9 

Corrected.

Reviewer’s question R1.10

Section II.A, paragraph 5, line 2: A reference number is missing,

Authors’ answer to R1.10 

Corrected

Reviewer’s question R1.11

Page 3, the bullet Single User MIMO: In latex, the left quotation mark should be ``. Fix all left quotation marks.

Authors’ answer to R1.11

Corrected

Reviewer’s question R1.12

Page 8, column 2, at end of paragraph 2: An MSE of position should have a unit of cm^2, not cm.

Authors’ answer to R1.12 

Corrected

Reviewer’s question R1.13

Page 8, column 2, paragraph 3: Change "(2.5m2.5m)" to "2.5m \times 2.5m"

Authors’ answer to R1.13 

Corrected

Reviewer’s question R1.14

Capitalize table/figure/section, e.g., at the beginning of Section IV, table I => Table I, figure 1 => Figure 1, sections: => Sections. 

Authors’ answer to R1.14 

All suggestions have been made in the revised version.

Reviewer’s question R1.15

Page 14, column 1, the last two lines: figure 2 and 3 => Figures 2 and 3

Authors’ answer to R1.15 

Corrected.

Reviewer 2 Report

  • The authors doing a good job to investigate this important field, as Deep Learning and Reinforcement Learning having great impact on mitigating issues in current MIMO communication systems, it is very valuable to write a survey that can be benefit to grasp future directions by reviewing current works.
  • The papers cited are very comprehensive and new, and the authors focus on hot spot application of DL and RL in MIMO systems, such as Channel Estimation, mmWave Communications, Positioning, Sensing, and Localization.
  • There is an unreasonable paragraph ordering problem here. The Part â…¥ in Page 17 introduces other related work and emphasizes the necessity of writing this paper. However, following the writing order, this part should better to be put at the front of the paper.
  • There is an unprecise description when introducing the background of MIMO communication. In Page 3 Section C, “Macrodiversity MIMO is a type of space diversity” may cause misunderstanding, as we all know, the space diversity is a diversity method belong to Micro-diversity which is contrary to Marcodiversity.
  • In Page 3 Section C, when introducing the background of MIMO, line 10, the use of “inference” is obviously incorrect. It is that the ordinary multi-path transmission without MIMO will contribute to multi-path interference, hence I wonder the author want to say “interference”.
  • In Page 14, the title of Fig. 1 need to be revised, as “ No of ” is completely incomprehensible.

Author Response

We are grateful to the Associate Editor and all the Reviewers for their valuable comments, which we have tried to integrate fully within this revised version of our paper. 

For details, please see our replies to the Reviewers’ comments.

Detailed responses to the Reviewer-2 questions are given below:

Reviewer’s general comment 

As a survey about the application of DL and RL in MIMO systems, this paper mainly presents a comprehensive review on the integration between those two areas. First, the paper well introduces the background of DL, RL and MIMO in Part â…¡, including a brief introduction and a classification of each one. Then the paper focuses on the applications of RL and DL to different issues of MIMO communication systems by illustrating seven aspects, the content discussed and the articles cited are very comprehensive, current and in-depth. After that, the authors present statistics about the related fields papers to demonstrate the impact and trend. Besides, the paper discusses the surveyed papers by highlighting the strengths and limitations, along with some future directions. The authors also investigate other related survey papers, they give some comments and list a table. With other related works can’t well cover the content about DL, RL and their application in MIMO, the necessity of this paper can be seen

Reviewer’s question R2.1

There is an unreasonable paragraph ordering problem here. The Part â…¥ in Page 17 introduces other related work and emphasizes the necessity of writing this paper. However, following the writing order, this part should better to be put at the front of the paper

Authors’ answer to R2.1

 Thanks for pointing out this issue. We have moved related work from section 6 to section 3.

Reviewer’s question R2.2

There is an unprecise description when introducing the background of MIMO communication. In Page 3 Section C, “Macrodiversity MIMO is a type of space diversity” may cause misunderstanding, as we all know, space diversity is a diversity method belong to Microdiversity which is contrary to Marcodiversity. 

Authors’ answer to R2.2 

We have modified the sentence highlighted  by the reviewer in order to avoid any confusion for the readers.

Reviewer’s question R2.3

In Page 13, the last paragraph of Part F where the author says the [259] proposed three NNs, however only two of them are be demonstrated here.

Authors’ answer to R2.3 

It is a significant observation from the reviewer and we have added the missing part in the revised version. Please note that the reference (ref [259]) will be changed in the revised version as we have moved the related work part to section 3.

Reviewer’s question R2.4

In Page 3 Section C, when introducing the background of MIMO, line 10, the use of “inference” is obviously incorrect. It is that the ordinary multi-path transmission without MIMO will contribute to multi-path interference, hence I wonder the author want to say “interference”

Authors’ answer to R2.4 

Yes, it was a typo mistake and we have corrected it. Thank you for the careful reading of the paper.

Reviewer’s question R2.5

In Page 14, the title of Fig. 1 needs to be revised, as “ No of ” is completely incomprehensible.

Authors’ answer to R2.5 

A correction has been made as advised by the reviewer.

Reviewer 3 Report

This paper provided a comprehensive survey on the applications of reinforcement learning and deep learning in multiple-input and multiple-output systems. The contribution and the presentation are well. Some of my comments are given as below to further improve this paper.

(1) There are some format problems such as: is given in[228].--> is given in [228]. ([9], [10], [11] and [12]),-> ([9-12]), You can combine all of these references together use format [9]-[12]. Please correct all of these references format problems in the whole paper. Also, there are some format problems after a paragraph, there is two space to start the next paragraph. 

(2) Reference format problems: Please check all of references [12] Y. Xin, “”machine learning and deep learning methods for cybersecurity”,  ieee access, vol. 6, pp. 35365-35381, 2018.,” should be corrected as: [12] Y. Xin, et al., “Machine learning and deep learning methods for cybersecurity,” IEEE Access, vol. 6, pp. 35365-35381, 2018.”

(3) The reference format: J. Wang, G. Gui, T. Ohtsuki, B. Adebisi, H. Gacanin, and H. Sari, “Compressive sampled csi feedback method based on deep learning for fdd massive mimo systems,” IEEE Transactions on Communications, 69(9): 5873-5885, 2021.

(4) Some related works are suggested to discuss in this paper. such as:

W. Ma, C. Qi, Z. Zhang and J. Cheng, ``Sparse channel estimation and hybrid precoding using deep learning for millimeter wave massive MIMO,'' IEEE Trans. Commun., vol. 68, no. 5, pp. 2838--2849, May 2020.

G. Fan, J. Sun, B. Adebisi, T. Ohtsuki, G. Gui,  H. Gacanin, and H. Sari, ``Limited CSI feedback using fully convolutional neural networks for FDD massive MIMO systems,'' IEEE Trans. Cogn. Commun. Netw., early access, doi: 10.1109/TCCN.2021.3119945

Z.~Wang, ``An adaptive deep learning-based UAV receiver design for coded MIMO with correlated noise,'' Physical Communication vol.~45, no. 101292, pp. 1--8, 2021.

Author Response

We are grateful to the Associate Editor and all the Reviewers for their valuable comments, which we have tried to integrate fully within this revised version of our paper. 

For details, please see our replies to the Reviewers’ comments.

Detailed responses to the Reviewer-3 questions are given below:

Reviewer’s general comment 

This paper provided a comprehensive survey on the applications of reinforcement learning and deep learning in multiple-input and multiple-output systems. The contribution and the presentation are well. Some of my comments are given as below to further improve this paper.

Reviewer’s question R3.1

There are some format problems such as: is given in[228].--> is given in [228]. ([9], [10], [11] and [12]),-> ([9-12]), You can combine all of these references together using format [9]-[12]. Please correct all of these reference format problems in the whole paper. Also, there are some format problems after a paragraph, there are two spaces to start the next paragraph.

Authors’ answer to R3.1

 We are thankful to the reviewers for highlighting this issue. We have revised all the references and have corrected the format as advised by the reviewer.

Reviewer’s question R3.2

Reference format problems: Please check all of references [12] Y. Xin, “”machine learning and deep learning methods for cybersecurity”,  ieee access, vol. 6, pp. 35365-35381, 2018.,” should be corrected as: [12] Y. Xin, et al., “Machine learning and deep learning methods for cybersecurity,” IEEE Access, vol. 6, pp. 35365-35381, 2018.”

Authors’ answer to R3.2 

We have used standard bibliography style through google scholar. It was a latex format that displayed some mistakes. Now we have used MDPI format and ieee access appears as IEEE Access.

Reviewer’s question R3.3

The reference format: J. Wang, G. Gui, T. Ohtsuki, B. Adebisi, H. Gacanin, and H. Sari, “Compressive sampled csi feedback method based on deep learning for fdd massive mimo systems,” IEEE Transactions on Communications, 69(9): 5873-5885, 2021

Authors’ answer to R3.3 

We have added volume and page information in the highlighted reference.

Reviewer’s question R3.4

Some related works are suggested to discuss in this paper. such as:

  1. Ma, C. Qi, Z. Zhang and J. Cheng, ``Sparse channel estimation and hybrid precoding using deep learning for millimeter wave massive MIMO,'' IEEE Trans. Commun., vol. 68, no. 5, pp. 2838--2849, May 2020.
  2. Fan, J. Sun, B. Adebisi, T. Ohtsuki, G. Gui,  H. Gacanin, and H. Sari, ``Limited CSI feedback using fully convolutional neural networks for FDD massive MIMO systems,'' IEEE Trans. Cogn. Commun. Netw., early access, doi: 10.1109/TCCN.2021.3119945

Z.~Wang, ``An adaptive deep learning-based UAV receiver design for coded MIMO with correlated noise,'' Physical Communication vol.~45, no. 101292, pp. 1--8, 2021

Authors’ answer to R3.4 

The first paper suggested by the reviewer is already discussed in the paper. Please see the ref [127] on the old version and ref [143] section 4.2, in the revised version. A discussion has been added on the second and third suggested papers. However, the title of the second paper is ‘’Fully Convolutional Neural Network Based CSI Limited Feedback for FDD Massive MIMO Systems’’ but the author's list and journal information is the same. We were unable to find the paper with the title as suggested by the reviewer.

Reviewer 4 Report

This survey paper describes Application of Reinforcement Learning and Deep Learning in Multiple-Input and Multiple-Output (MIMO) Systems.

I have the following recommendations regarding the improvement of the paper.

Add some recent literature published in the last 3 years to the paper. 

Change the format to mdpi sensors standard format available on the sensors mdpi website.

MIMO acrnonym should be defined in the abstract and introduction parat on its first appearance.

Add a figure showing how RL works. Similarly for MIMO add a concept figure.

Describe the limitations of your own work in Table III.

What is the contribution of AI in Table III. 

Regarding ML limitation in Table III. illustrate limitation of ML in detail.

Add three more applications to table II.

Author Response

We are grateful to the Associate Editor and all the Reviewers for their valuable comments, which we have tried to integrate fully within this revised version of our paper. 

For details, please see our replies to the Reviewers’ comments.

Detailed responses to the Reviewer-4 questions are given below:

Reviewer’s general comment 

This survey paper describes Application of Reinforcement Learning and Deep Learning in Multiple-Input and Multiple-Output (MIMO) Systems.

I have the following recommendations regarding the improvement of the paper.

Reviewer’s question R4.1

Add some recent literature published in the last 3 years to the paper.

Authors’ answer to R4.1

 We have made a quick search in available time and we have added the following papers on the advice of the reviewer.

  1. Fan, J. Sun, B. Adebisi, T. Ohtsuki, G. Gui,  H. Gacanin, and H. Sari, ``Limited CSI feedback using fully convolutional neural networks for FDD massive MIMO systems,'' IEEE Trans. Cogn. Commun. Netw., early access, doi: 10.1109/TCCN.2021.3119945

Z.~Wang, ``An adaptive deep learning-based UAV receiver design for coded MIMO with correlated noise,'' Physical Communication vol.~45, no. 101292, pp. 1--8, 2021

Han, Yu and Li, Mengyuan and Jin, Shi and Wen, Chao-Kai and Ma, Xiaoli, Deep learning-based FDD non-stationary massive MIMO downlink channel reconstruction, IEEE Journal on Selected Areas in Communications, 2020.

Lu, Qiujin and Lin, Tian and Zhu, Yu, Channel Estimation and Hybrid Precoding for Millimeter Wave Communications: A Deep Learning-Based Approach,  IEEE Access, 2021

Song, Nuan and Ye, Chenhui and Hu, Xiaofeng and Yang, Tao, Deep Learning based Low-Rank Channel Recovery for Hybrid Beamforming in Millimeter-Wave Massive MIMO,  IEEE Wireless Communications and Networking Conference (WCNC), 2020.

Reviewer’s question R4.2

Change the format to mdpi sensors standard format available on the sensors mdpi website

Authors’ answer to R4.2 

We have changed the format and the revised version is in MDPI format.

Reviewer’s question R4.3

MIMO acrnonym should be defined in the abstract and introduction parat on its first appearance.

Authors’ answer to R4.3 

We have made the suggested changes.

Reviewer’s question R4.4

Add a figure showing how RL works. Similarly for MIMO add a concept figure

Authors’ answer to R4.4 

We agree with the reviewer that figures for both technologies were missing. We have added corresponding figures in the revised manuscript. Please see figure 1 and figure 2.

Reviewer’s question R4.5 

Describe the limitations of your own work in Table III.

Authors’ answer to R4.5 

An apparent limitation has been added in the revised version on the reviewer's advice.

Reviewer’s question R4.6

What is the contribution of AI in Table III.

Authors’ answer to R4.6 

Contribution of the highlighted paper was mistakenly missing. We have added the missing text in the revised version. Thanks for pointing out this mistake.

Reviewer’s question R4.7

Regarding ML limitation in Table III. illustrate the limitation of ML in detail.

Authors’ answer to R4.7

In the related work table (Table 3 in the old version), our intention was to highlight the limitation of existing survey papers regarding application of RL and DL in MIMO systems. Therefore, for the highlighted paper, the limitation from our point of view is that it concentrates on generic ML techniques instead of RL and DL. Highlighting the limitation of ML is out of the scope of our paper.

Reviewer’s question R4.8

Add three more applications to table II.

Authors’ answer to R4.8 

Table II (now Table 3 in the revised version) was created as the result of the Application section. More precisely, each row in Table 3 is a subsection of the Application section and each subsection consists of a minimum of 2 pages with many cited papers. We have categorized the papers accordingly with the results of our survey. It is difficult for us to figure out new categories with respect to the set of papers we have found.